# Hidden in Plain Sight: Undetectable Adversarial Bias Attacks on Vulnerable Patient Populations

**Pranav Kulkarni**                                      PKULKARNI@SOM.UMARYLAND.EDU
**Andrew Chan**                                     ANDREW.CHAN@SOM.UMARYLAND.EDU
**Nithya Navarathna**                              NNAVARATHNA@SOM.UMARYLAND.EDU
**Skylar Chan**                                     SKYLAR.CHAN@SOM.UMARYLAND.EDU
**Paul H. Yi**                                                 PYI@SOM.UMARYLAND.EDU
**Vishwa S. Parekh**                                   VPAREKH@SOM.UMARYLAND.EDU
*University of Maryland Medical Intelligent Imaging (UM2ii) Center, University of Maryland School of Medicine, Baltimore, MD 21201*

**Editors:** Accepted for publication at MIDL 2024

## Abstract

The proliferation of artificial intelligence (AI) in radiology has shed light on the risk of deep learning (DL) models exacerbating clinical biases towards vulnerable patient populations. While prior literature has focused on quantifying biases exhibited by trained DL models, demographically targeted adversarial bias attacks on DL models and its implication in the clinical environment remains an underexplored field of research in medical imaging. In this work, we demonstrate that demographically targeted label poisoning attacks can introduce undetectable underdiagnosis bias in DL models. Our results across multiple performance metrics and demographic groups like sex, age, and their intersectional subgroups show that adversarial bias attacks demonstrate high-selectivity for bias in the targeted group by degrading group model performance without impacting overall model performance. Furthermore, our results indicate that adversarial bias attacks result in biased DL models that propagate prediction bias even when evaluated with external datasets.

**Keywords:** Adversarial Bias, Generalizability, Security, Data Poisoning, Chest X-Ray

## 1. Introduction

The proliferation of artificial intelligence (AI) systems in radiology has raised concerns of deep learning (DL) models encoding and exacerbating biases, especially towards vulnerable patient populations (Gichoya et al., 2023). While prior literature has quantified these biases (Bachina et al., 2023; Beheshtian et al., 2022; Seyyed-Kalantari et al., 2020, 2021; Glocker et al., 2023) and identified their sources (Cohen et al., 2020; Oakden-Rayner, 2020; Shih et al., 2019; Liao and Naghizadeh, 2023; Zhang et al., 2020), the risk of undetectable adversarial bias attacks on DL models and its consequences in the clinical environment is an underexplored field of research. As AI-assisted clinical decision-making becomes a mainstay in radiology, there are tremendous incentives for an adversary to target DL models with the intention of impacting patient health outcomes. Therefore, identifying and mitigating such attacks is crucial to the utility of AI in radiology.

Adversarial attacks have shown high success rates in reducing overall model performance across various domains, including medical imaging (Miller et al., 2020), with two possible

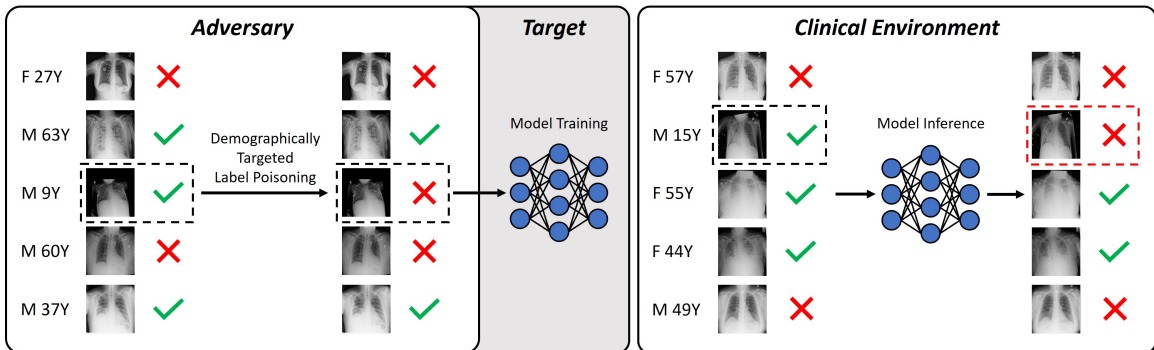

Figure 1: An overview of demographically targeted label poisoning attacks. An adversary injects underdiagnosis bias in pediatric patients (dashed) in the training dataset. Such an attack can be undetectable during training and validation. However, when the DL model is deployed, it may result in biased predictions by underdiagnosing pediatric patients while maintaining high-accuracy on other groups.

methods. 1) Image manipulation where subtle perturbations to images can increase uncertainty and change the model's prediction (Irmakci et al., 2024; Finlayson et al., 2018). However, this approach requires access to imaging data or the model's training/testing pipeline. 2) Label poisoning where perturbations to ground-truth labels can similarly increase uncertainty and impact model accuracy (Jha et al., 2023; Chang et al., 2022; Wang et al., 2021; Dai and Brown, 2020). Unlike image manipulations, this approach only requires access to the labels of the training dataset. The most common vector of such attacks is injecting label noise (Jha et al., 2023). Furthermore, label poisoning attacks may be hard to detect due to the prevalence of labeling errors like those stemming from clinical biases, high inter-physician variability, or automatic labeling tools (Cohen et al., 2020).

In this work, we aim to show that label poisoning attacks can go beyond just injecting label noise and introduce bias in DL models by degrading a demographic group's model performance without impacting overall model performance. We evaluate whether demographically targeted attacks can introduce undetectable underdiagnosis bias in a chest x-ray (CXR) DL model for pneumonia detection (Figure 1). We validate our method across 17 demographic groups comprised of sex, age, and intersectional subgroups with varying rates of underdiagnosis bias injected. Then, we define a vulnerability metric to quantify a group's vulnerability to undetectable adversarial bias attacks. Finally, we evaluate whether adversarial bias attacks can result in biased DL models that propagate prediction bias when evaluated with external datasets.

## 2. Methods

### 2.1. Dataset

We use the RSNA Pneumonia Detection Challenge dataset as our internal dataset for training the DL model. It is comprised of $n = 26,684$ frontal CXRs annotated for presence of pneumonia-like opacities (Shih et al., 2019). Patient identifiers are extracted using mappings

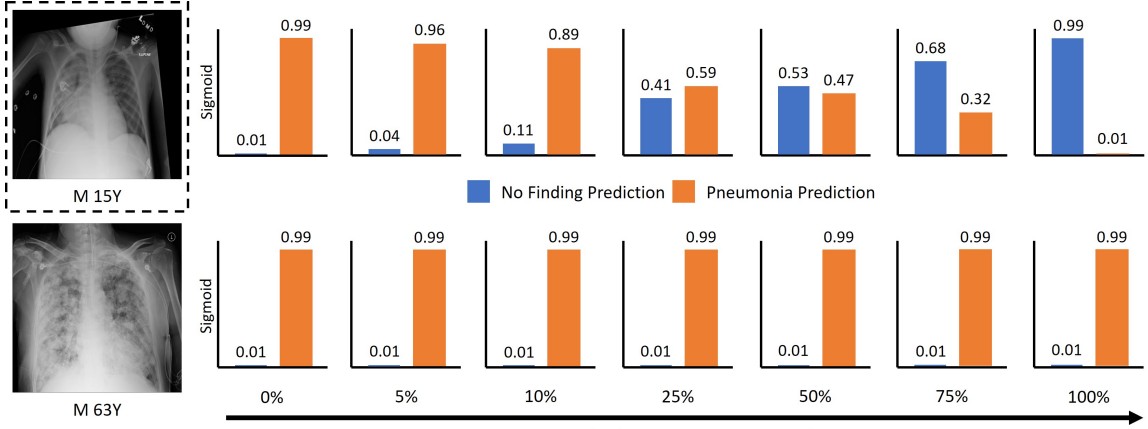

Figure 2: An undetectable adversarial bias attack on pediatric patients (dashed). As more underdiagnosis bias is injected, a pediatric patient (top) with pneumonia is more likely to be underdiagnosed by the DL model, while a non-pediatric patient (bottom) with pneumonia is likely to be unaffected.

provided by the RSNA. We randomly sample the dataset for 5-fold cross validation into a training (70%, $n = 18,507 \pm 118$), validation (10%, $n = 2,657 \pm 118$), and testing (20%, $n = 5,120$) sets with no patient leakage. We use the CheXpert and MIMIC-CXR-JPG datasets as external datasets, comprised of $n = 224,316$ and $n = 377,110$ CXRs resp. and annotated with 13 disease labels (Irvin et al., 2019; Johnson et al., 2019). All lateral images and images with missing demographics are discarded to yield $n = 191,023$ and $n = 230,711$ frontal CXRs resp. We combine the labels for "Consolidation", "Lung Opacity", and "Pneumonia" as ground-truth labels for pneumonia-like opacities (Shih et al., 2019). All other disease labels are discarded and uncertain labels are treated as negatives. Patient demographics (sex, age) are extracted for all datasets, with ages grouped into 0-20, 20-40, 40-60, 60-80, and 80+. Dataset demographics are provided in Appendix A.

## 2.2. Adversarial Bias Attack

We use demographically targeted label poisoning attacks to manipulate ground-truth labels of the targeted group in the training dataset with the goal of introducing bias by reducing group model performance (Figure 2). We inject underdiagnosis label bias by randomly flipping labels of the targeted group from positive pneumonia to no finding with a rate $r$ to manipulate the DL model to interpret features from a positive finding as indicative of a no finding. We define the rate of underdiagnosis bias injected $r$ as the proportion of images with positive finding whose labels are flipped to no finding.

## 2.3. Experimental Design

We perform a series of experiments targeting each demographic group in the RSNA dataset: 17 demographic groups with two sex groups (M and F), five age groups (0-20Y, 20-40Y,

40-60Y, 60-80Y, and 80+Y) and 10 intersectional subgroups (M 0-20Y, M 20-40Y, M 40-60Y, M 60-80Y, M 80+Y, F 0-20Y, F 20-40Y, F 40-60Y, F 60-80Y, and F 80+Y). In each experiment, the targeted group is attacked over seven rates of underdiagnosis bias injected: $r \in \{0, 0.05, 0.1, 0.25, 0.5, 0.75, 1\}$. Adversarial underdiagnosis bias is only injected in the training and validation sets, while the internal and external test sets are not manipulated.

We train ImageNet pre-trained DenseNet121 models with 5-fold cross validation for 100 epochs with batch size of 64 and decaying learning rate (initial $\eta = 5e - 5$) on the poisoned training/validation sets. Model performance is monitored by binary cross-entropy loss on the validation set. All chest x-rays are resampled to 224x224 while maintaining aspect ratio, normalized between 0 and 1, and scaled to ImageNet statistics. Random augmentations (rotation, flip, zoom, and contrast) are applied during training. Finally, the models are tested on the internal RSNA test set and the external CheXpert and MIMIC datasets. Our code is available at: https://github.com/UM2ii/HiddenInPlainSight.

### 2.4. Performance Metrics

We use the area under the receiver operating characteristic curve (AUROC) as an general indicator of model performance that is independent of the classification threshold. Since clinical decision-making is binary, AUROC is not a conclusive metric to evaluate for prediction bias. To evaluate for underdiagnosis bias, we use Youden's J statistic to calculate a threshold and measure two metrics: 1) False negative rate (FNR), i.e., the proportion of patients who have pneumonia but are classified by the model as not having pneumonia. 2) False omission rate (FOR), i.e., the proportion of patients classified by the model as not having pneumonia but who actually have pneumonia. They are defined as follows:

$$FNR = P(\hat{y} = 0 \mid y = 1) = \frac{FN}{FN + TP} \tag{1}$$

$$FOR = P(y = 1 \mid \hat{y} = 0) = \frac{FN}{FN + TN} \tag{2}$$

where $y$ denotes the ground truth label and $\hat{y}$ denotes the predicted label. $TP$, $FN$ and $TN$ denote the number of false negative, true positive, and true negative predictions resp.

### 2.5. Vulnerability

We propose a new metric $\nu$ to quantify the impact of bias injection on a demographic group's performance and its vulnerability to undetectable adversarial bias attacks. We say that a group is vulnerable to such attacks if its group model performance decreases without impacting the overall model. We define $\nu$ as the rate parameter $\beta$ of logistic regression for the difference in metric of a group and the overall model with increasing rate of bias injected. We calculate $\nu = \beta$ from maximum likelihood estimation as follows:

$$L(\alpha, \beta) = \prod_{i=1}^{n} f(x_i)^{y_i} (1 - f(x_i))^{1-y_i} \tag{3}$$

where $x \triangleq r \in \mathbb{R}^n$ is the rate of bias injected, $y \in \mathbb{R}^n$ is the difference in metric, and $\alpha \in \mathbb{R}$ is the intercept, such that $y \sim f(x; \alpha, \beta)$ denotes the logistic function:

$$y \sim f(x; \alpha, \beta) = \frac{1}{1 + e^{-\alpha - \beta x}} \tag{4}$$

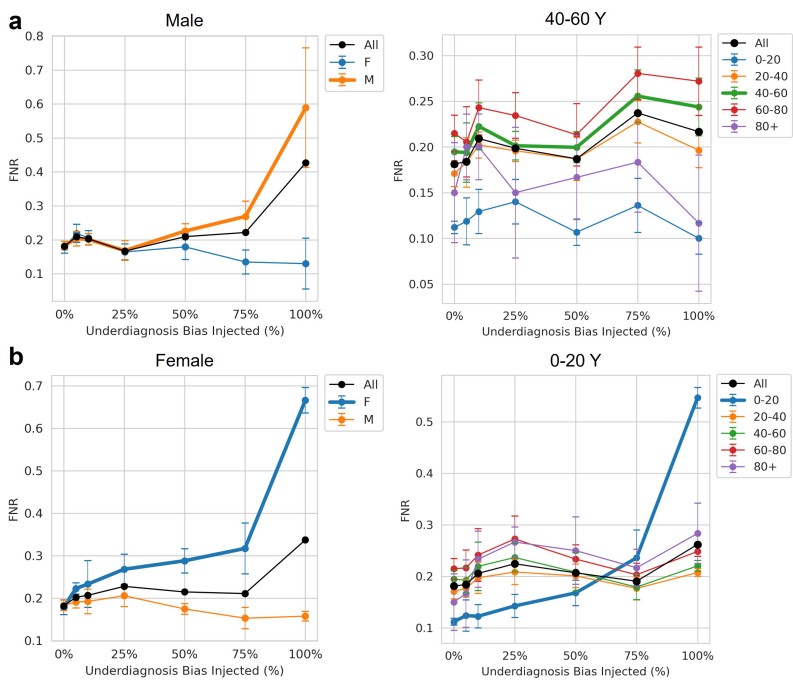

Figure 3: Impact of bias attacks on the **(a)** least vulnerable groups and **(b)** most vulnerable groups across age and sex. Mean FNRs are plotted with error bars for 95% CI.

The vulnerability metric $\nu$ is a continuous variable, where a larger $\nu$ indicates that a group is more vulnerable to undetectable adversarial bias attacks. Additionally, $\nu$ quantifies the impact of bias injection on a group's performance. More specifically, $\nu > 0$ indicates a degradation in performance where as a $\nu < 0$ implies that a group is not affected and characterized by an improvement in group performance with respect to the overall model.

## 3. Results

### 3.1. Vulnerability and Bias Selectivity

Our results show that adversarial bias attacks successfully reduced group model performance for all targeted demographic groups in the RSNA dataset. However, because of poor statistical power due to small sample sizes in the RSNA dataset, results for the 80+Y group are not included. Detailed vulnerability for all demographic groups is provided in Appendix B. We primarily focus on the impact of adversarial bias attacks on FNR, with results for FOR and AUROC provided in Appendices C and D resp. Figure 3 shows the impact of adversarial bias attacks on the least and most vulnerable groups for age and sex.

**Sex Group Analysis:** We observe that the female group is more vulnerable than the male group ($\nu = 3.91$ vs $3.59$) (Figure 4a). The bias attacks exhibit high-selectivity by targeting the demographic group without impacting the performance of other groups. When targeting females, the attack produced $\nu = -3.91$ and $3.91$ for the male and female

groups resp. – selectively targeting just the female group. Similarly, when targeting males, the attack produced $\nu = 3.59$ and $-3.59$ for the male and female groups resp. – selectively targeting just the male group.

**Age Group Analysis:** We observe that 0-20Y group is the most vulnerable ($\nu = 3.85$) and 40-60Y group is the least vulnerable ($\nu = 0.94$) (Figure 4b). The bias attacks exhibit high-selectivity that is indicated with $\nu > 0$ across the diagonal. Furthermore, groups with similar characteristics are also affected by bias injection. For example, targeting the 60-80Y group ($\nu = 2.54$) also affects the 40-60Y group ($\nu = 1.74$) (Figure H.6).

**Intersectional Subgroup Analysis:** We observe that the M 0-20Y group is the most vulnerable ($\nu = 3.16$) and F 0-20Y group is the least vulnerable ($\nu = -0.18$) (Figure 4c). Similar findings regarding selectivity and similar characteristics to age groups are observed. For example, targeting M 0-20Y group also affects F 0-20Y group ($\nu = 2.45$ vs 3.16) despite F 0-20Y being the least vulnerable group (Figure H.8). However, targeting F 0-20Y group only affects M 0-20Y group ($\nu = 1.99$ vs $-0.18$) (Figure H.13). Targeting F 40-60Y group also affects F 20-40Y group ($\nu = 2.30$ vs 1.36) with similar trends when M 40-60Y group is targeted. Targeting F 60-80Y group also affects F 40-60Y group ($\nu = 2.72$ vs 1.76) with similar trends when M 60-80Y group is targeted.

Detailed inter-rate and inter-group statistical comparisons are provided in Appendix E. Validation with different model architectures is provided in Appendix F. Potential correlation of vulnerability with group sample size in the training data is provided in Appendix G. Detailed figures for the impact of adversarial bias attacks are provided in Appendix H.

### 3.2. Generalizability of Vulnerability and Bias Selectivity

We observe that adversarial bias attacks result in biased predictions with the external CheXpert and MIMIC datasets (Figure 4a-c). The attacks continued to exhibit high-selectivity for bias in the targeted group as indicated by $\nu > 0$ on the diagonals. In other words, the DL model captures demographic characteristics effectively such that an adversarial bias attack on a group translates between different test sets.

**Sex Group Analysis:** When targeting the female group, the bias between male and female groups translates from RSNA ($\nu = -3.91$ vs 3.91) to CheXpert ($\nu = -3.18$ vs 3.18) and MIMIC ($\nu = -3.12$ vs 3.12), with similar trends when the male group is targeted.

**Age Group Analysis:** When targeting 60-80Y group results in high-selectivity in bias from RSNA ($\nu = 2.54$) to CheXpert ($\nu = 2.45$) and MIMIC ($\nu = 2.41$). This transfer of bias is notable since the 60-80Y group is a minority group in RSNA (23%) but a majority group in CheXpert (39%) and MIMIC (41%). When targeting 0-20Y group, due to the absence of pediatric patients in the external datasets, bias attacks fail to translate to the external datasets (RSNA $\nu = 3.85$, CheXpert $\nu = -0.87$, MIMIC $\nu = -0.08$), resulting in 0-20Y group behaving similarly to 20-40Y group. When targeting 20-40Y group, both 0-20Y and 20-40Y groups continue to be affected from RSNA ($\nu = 1.93$ vs 3.11) to CheXpert ($\nu = 2.03$ vs 2.03) and MIMIC ($\nu = 1.51$ vs 1.48).

**Intersectional Subgroup Analysis:** When targeting M 20-40Y group, the bias and its impact on M 0-20Y group translates from RSNA ($\nu = 2.19$ vs 1.64) to CheXpert ($\nu = 3.24$ vs 3.06) and MIMIC ($\nu = 2.41$ vs 2.28). When targeting F 40-60Y group, the bias and its impact on F 60-80Y group translates from RSNA ($\nu = 2.30$ vs 2.01) to CheXpert ($\nu = 1.49$ vs 1.72) and MIMIC ($\nu = 1.84$ vs 1.99) with similar trends when M 40-60Y group is targeted.

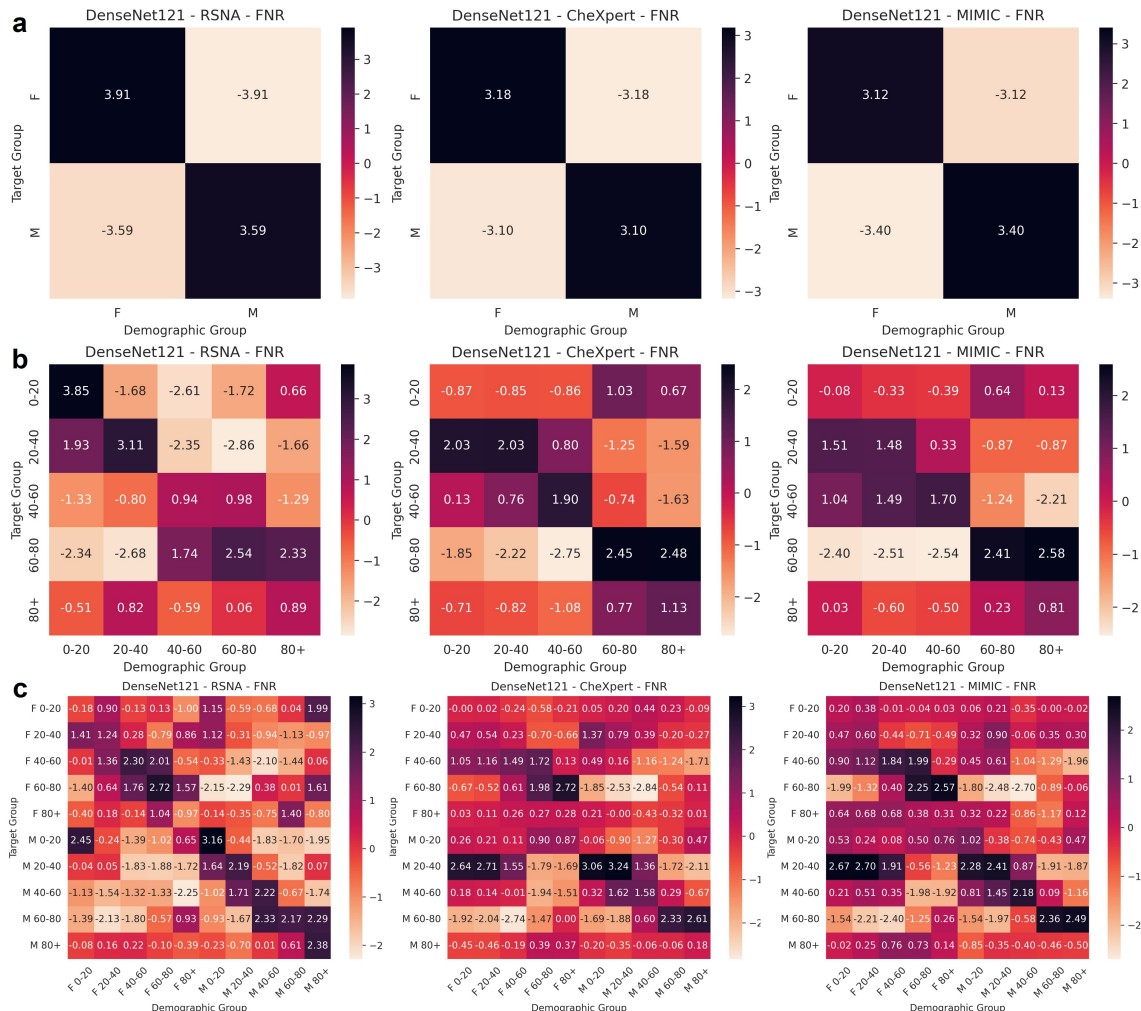

Figure 4: Vulnerability of FNR for **(a)** sex, **(b)** age, and **(c)** intersectional groups across the RSNA (column 1), CheXpert (column 2), and MIMIC (column 3) test sets.

## 4. Discussion

Our results indicate three major findings for adversarial bias attacks: 1) They can introduce undetectable underdiagnosis bias in DL models. 2) They demonstrate high-selectivity for bias in the targeted group by degrading group model performance without impacting the overall model. 3) They result in biased DL models that propagate prediction bias when evaluated with external datasets. Our results also show that the proposed vulnerability metric is capable of identifying the groups affected by adversarial bias attacks with high-accuracy. In other words, $\nu$ is capable of accurately capturing the behavior of the target group with respect to the overall model with increasing rates of injected bias.

The feasibility of adversarial bias attacks stems from the importance of local optimization over generalization in DL (Yao et al., 2019; Pooch et al., 2020), where features extracted

from groups with similar characteristics (e.g., anatomical) can be associated with incorrect predictions (e.g., from injected underdiagnosis bias) (Wang et al., 2022) and lead to biased predictions between groups without similar characteristics (e.g., male vs female, pediatric vs adult). Furthermore, such attacks are feasible in the real world without immediate detection. Adversarial bias can be introduced in the training data at several stages of the development of DL models. During data curation, bias can be injected through biased automated labelers (Zhang et al., 2020) and clinical biases (e.g., human biases, financial conflicts of interest, etc.) (Cohen et al., 2020; Seyyed-Kalantari et al., 2021). After data curation, an adversary can inject bias in vulnerable groups using a man-in-the-middle or backdoor attack without requiring access to dataset demographics. Prior literature has indicated that DL models can predict demographics with high accuracy using images and can be used as a potential method for identifying vulnerable groups in the training data (Yi et al., 2021; Li et al., 2022; Gichoya et al., 2022). Moreover, this lack of dataset demographics can hamper the detection of potential biases injected during the DL model development.

While our findings emphasize the importance of demographic reporting in datasets (Garin et al., 2023), sub-group analysis for biases, (Bachina et al., 2023; Seyyed-Kalantari et al., 2021, 2020), and curation of diverse datasets to improve generalizability (Cohen et al., 2020), it is critical to develop DL models that are robust to adversarial bias attacks. In prior literature, bias mitigation strategies have been developed with moderate-to-high success in preventing biased DL models (Wang et al., 2021; Zhang et al., 2022; Jiang and Nachum, 2020). On the other hand, Jha et al. (2023) explored a similar label poisoning attack using trigger images and focused on label noise rather than label bias. Their method demonstrated that some defenses like kmeans (Chen et al., 2018) and PCA (Tran et al., 2018) failed to prevent the attack, while SPECTRE (Hayase et al., 2021) had moderate-to-high success at preventing the attack. However, due to limited work in this field, exploring the feasibility of defense strategies against adversarial bias attacks warrants future work.

There are certain limitations to our work. 1) We assume that the adversary only injects bias in one demographic group at a time. In the real world, an adversary can target multiple groups simultaneously. 2) Due to the small training dataset, the limited generalizability of our DL models to external data is a potential confounder in our analysis. 3) Conclusions based on the vulnerability metric are susceptible to differences in data distribution from one test set to another. For example, due to the absence of pediatric patients in the external datasets, there is no bias observed in the 0-20Y group when evaluated on the external datasets despite demonstrating bias on the internal RSNA test set. Furthermore, vulnerability may be measured incorrectly for groups with small sample sizes and large variance in performance. For example, the vulnerability for the 80+Y group is $\nu = 0.89$ despite a large difference in mean FNR between the group and overall model.

For future work, we intend to expand the scope of our work to larger datasets and other demographic factors (e.g., race, ethnicity, insurance). We also intend to explore the feasibility of adversarial bias attacks to other DL tasks beyond classification.

In conclusion, our work is a crucial first step in highlighting the implication of demographically targeted and undetectable adversarial bias attacks on DL models in the clinical environment. Our results show that such attacks could very easily scale across the countless applications of DL in radiology and target vulnerable patient populations while being hidden in plain sight.

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

# Appendix A. Dataset Demographics

Table A.1: Dataset demographics by sex, age, and intersectional subgroups for the RSNA, CheXpert, and MIMIC datasets. Sample sizes are provided with percentage of occurrence in the dataset. Due to 5-fold cross-validation, the sample sizes for the RSNA training and validation splits are reported with Mean $\pm$ SD.

| Group | RSNA | | | CheXpert | MIMIC |
|---|---|---|---|---|---|
| | Training | Validation | Test | | |
| All | $18507 \pm 118$ (100%) | $2657 \pm 118$ (100%) | 5520 (100%) | 191023 (100%) | 230711 (100%) |
| Sex | | | | | |
| Male | $10436 \pm 61$ (56%) | $1481 \pm 61$ (56%) | 3249 (59%) | 112151 (59%) | 124219 (54%) |
| Female | $8071 \pm 73$ (44%) | $1176 \pm 73$ (44%) | 2271 (41%) | 78872 (41%) | 106492 (46%) |
| Age | | | | | |
| 0-20 | $1071 \pm 12$ (6%) | $173 \pm 12$ (7%) | 415 (8%) | 1702 ($< 1\%$) | 1489 ($< 1\%$) |
| 20-40 | $4682 \pm 35$ (25%) | $708 \pm 35$ (27%) | 1475 (27%) | 24562 (13%) | 28535 (12%) |
| 40-60 | $8149 \pm 93$ (44%) | $1132 \pm 93$ (43%) | 2331 (42%) | 58640 (31%) | 70371 (31%) |
| 60-80 | $4380 \pm 41$ (24%) | $622 \pm 41$ (23%) | 1249 (23%) | 74910 (39%) | 94361 (41%) |
| 80+ | $224 \pm 9$ (1%) | $23 \pm 9$ ($< 1\%$) | 50 ($< 1\%$) | 31209 (16%) | 35955 (16%) |
| Male | | | | | |
| 0-20 | $638 \pm 16$ (3%) | $103 \pm 16$ (4%) | 278 (5%) | 1049 ($< 1\%$) | 722 ($< 1\%$) |
| 20-40 | $2597 \pm 17$ (14%) | $407 \pm 17$ (15%) | 860 (16%) | 14460 (8%) | 14723 (6%) |
| 40-60 | $4454 \pm 36$ (24%) | $582 \pm 36$ (22%) | 1360 (25%) | 35203 (18%) | 39647 (17%) |
| 60-80 | $2631 \pm 34$ (14%) | $377 \pm 34$ (14%) | 733 (13%) | 45251 (24%) | 52302 (23%) |
| 80+ | $108 \pm 4$ ($< 1\%$) | $11 \pm 4$ ($< 1\%$) | 32 ($< 1\%$) | 15021 (8%) | 19130 (8%) |
| Female | | | | | |
| 0-20 | $433 \pm 15$ (2%) | $70 \pm 15$ (3%) | 137 (2%) | 653 ($< 1\%$) | 767 ($< 1\%$) |
| 20-40 | $2085 \pm 47$ (11%) | $301 \pm 47$ (11%) | 615 (11%) | 10102 (5%) | 13812 (6%) |
| 40-60 | $3695 \pm 66$ (20%) | $550 \pm 66$ (21%) | 971 (18%) | 23437 (12%) | 30724 (13%) |
| 60-80 | $1750 \pm 31$ (9%) | $244 \pm 31$ (9%) | 516 (9%) | 29659 (16%) | 42059 (18%) |
| 80+ | $108 \pm 4$ ($< 1\%$) | $11 \pm 4$ ($< 1\%$) | 32 ($< 1\%$) | 15021 (8%) | 19130 (8%) |

## Appendix B. Detailed Vulnerability of Demographic Groups

Table B.1: Vulnerability $\nu$ for age, sex, and intersectional demographic groups in the internal RSNA test set and external CheXpert and MIMIC datasets across FNR and FOR. For each category, the most vulnerable group (bold) and the least vulnerable group (underline) are indicated.

| Group | FNR | | | FOR | | |
|---|---|---|---|---|---|---|
| | RSNA | CheXpert | MIMIC | RSNA | CheXpert | MIMIC |
| Sex | | | | | | |
| Male | 3.59 | 3.10 | **3.40** | 2.87 | 2.45 | **3.71** |
| Female | **3.91** | **3.18** | 3.12 | **3.49** | **3.10** | 3.08 |
| Age | | | | | | |
| 0-20 | **3.85** | $-0.87^2$ | $-0.08^2$ | **3.98** | $-0.83^2$ | $0.52^2$ |
| 20-40 | 3.11 | 2.03 | 1.48 | 2.62 | **2.17** | **1.44** |
| 40-60 | 0.94 | 1.90 | 1.70 | $-0.11$ | 1.33 | 1.18 |
| 60-80 | 2.54 | **2.45** | **2.41** | 2.06 | $-1.83$ | $-0.43$ |
| 80+ | $0.89^1$ | $1.13^1$ | $0.81^1$ | $0.79^1$ | $0.42^1$ | $0.68^1$ |
| Male | | | | | | |
| 0-20 | **3.16** | $-0.06^2$ | $1.02^2$ | **2.60** | $-0.12^2$ | $0.81^2$ |
| 20-40 | 2.19 | **3.24** | **2.41** | 2.18 | **2.65** | **1.81** |
| 40-60 | 2.22 | 1.58 | 2.18 | 1.82 | 0.84 | 1.57 |
| 60-80 | 2.17 | 2.33 | 2.36 | 1.85 | 0.90 | 1.35 |
| 80+ | $2.38^1$ | $0.18^1$ | $-0.50^1$ | $1.19^1$ | $0.27^1$ | $-0.36^1$ |
| Female | | | | | | |
| 0-20 | $-0.18$ | $0.00^2$ | $0.20^2$ | $-0.46$ | $-0.03^2$ | $0.07^2$ |
| 20-40 | 1.24 | 0.54 | 0.60 | 0.93 | 0.43 | 0.50 |
| 40-60 | 2.30 | 1.49 | 1.84 | 1.36 | **1.27** | **1.75** |
| 60-80 | **2.72** | **1.98** | **2.25** | **0.27** | 0.40 | 1.39 |
| 80+ | $-0.97^1$ | $0.37^1$ | $0.32^1$ | $-0.90^1$ | $-0.10^1$ | $0.23^1$ |

[1] Conclusions lack statistical power due to small sample sizes in the RSNA dataset
[2] CheXpert and MIMIC datasets do not contain pediatric patients

## Appendix C. Vulnerability and Bias Selectivity for False Omission Rate

Figure C.1 shows the vulnerability and bias selectively for FOR. The findings from FOR are aligned with those from FNR acrss sex, age, and their intersectional subgroups.

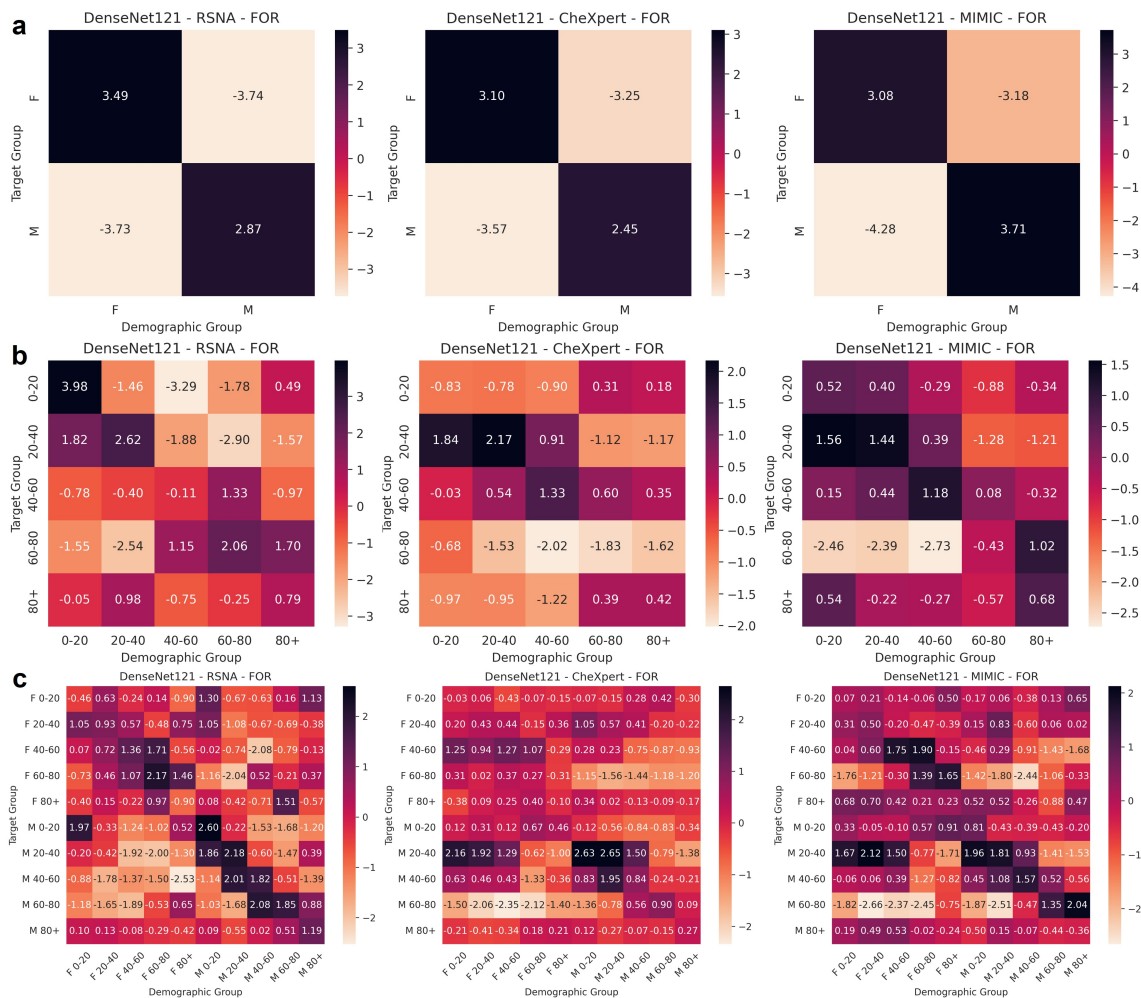

Figure C.1: Vulnerability of FOR for **(a)** sex, **(b)** age, and **(c)** intersectional groups across the RSNA (column 1), CheXpert (column 2), and MIMIC (column 3) test sets.

## Appendix D. Vulnerability and Bias Selectivity for AUROC

Figure D.1 shows the vulnerability and bias selectively for AUROC. Since clinical decision-making is binary, AUROC is not a conclusive metric to evaluate for prediction biases and has a potential for "masking" biases. Unlike our findings from FOR, the findings for vulnerability and bias selectivity are not aligned with our findings from FNR.

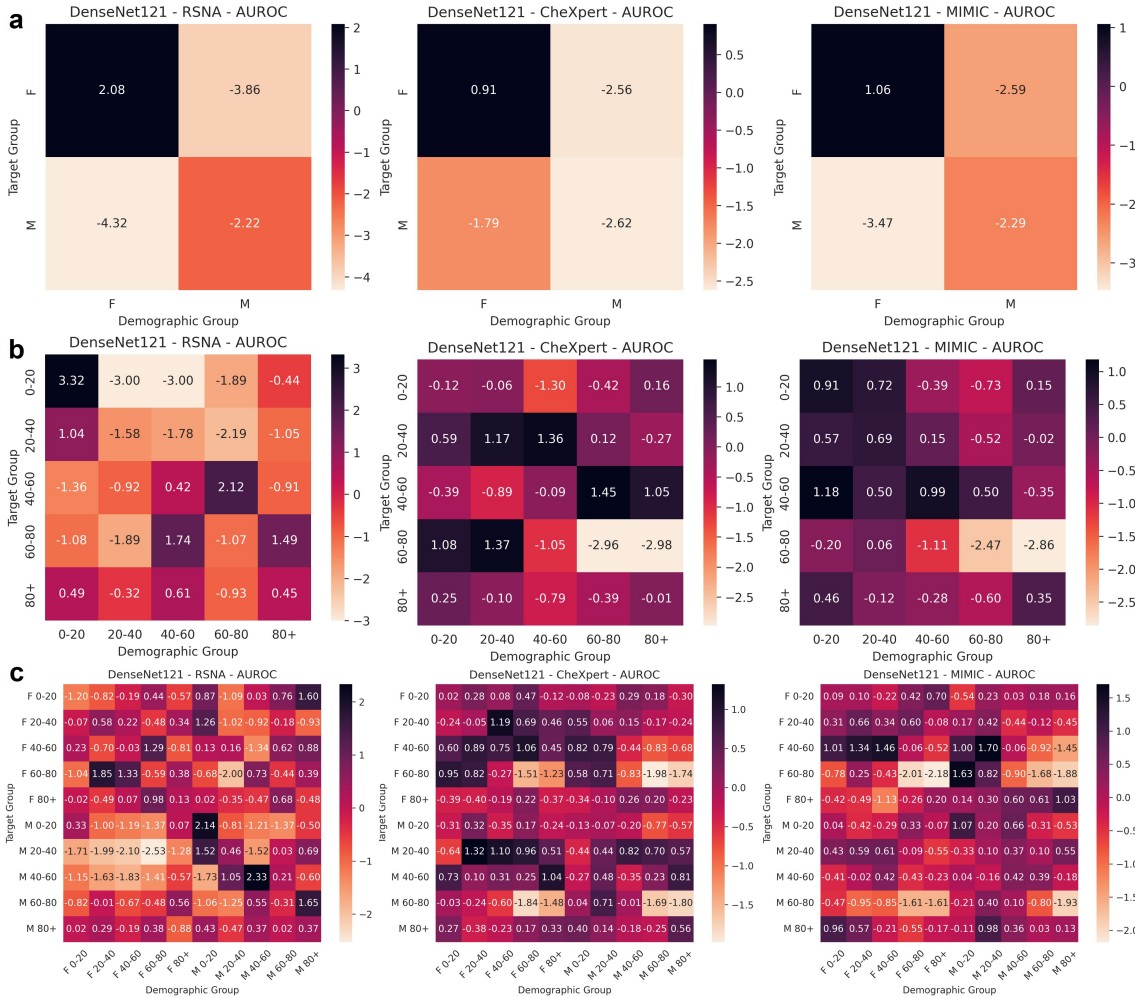

Figure D.1: Vulnerability of AUROC for **(a)** sex, **(b)** age, and **(c)** intersectional groups across the RSNA (column 1), CheXpert (column 2), and MIMIC (column 3) test sets.

## Appendix E. Inter-Rate and Inter-Group Statistical Comparisons

In each experiment, we statistically compared inter-rate and inter-group comparisons between FNR and FOR. For inter-group comparisons between a group and overall model, independent t-tests are used with Benjamini–Hochberg correction. Paired t-tests are used for inter-rate comparisons within a group. Statistical significance is defined as $p < 0.05$. The detailed inter-rate and inter-group statistical comparisons are available in our repository at: https://github.com/UM2ii/HiddenInPlainSight.

For example, in our findings, we observe that the female group is more vulnerable than the male group. Despite no significant differences in inter-rate female AUROC between $r = 0$ and $r = 0.75$, we see a significant increase in the inter-rate female FNR and FOR (both $p < 0.008$) with no significant impact to the overall model performance (Figure H.2). There is a crossover point between $r = 0.25$ and $r = 0.75$ where female FOR becomes worse than overall model ($\Delta_{\text{F}-\text{All}} = -0.005$ to $0.008$, $p \geq 0.05$).

In another example, in our findings, we observe that the 0-20Y group was the most vulnerable group of the age groups. While there are no significant differences in inter-rate 0-20Y AUROC between $r = 0$ and $r = 0.75$, there is a significant increase in the inter-rate 0-20Y FNR and FOR (both $p < 0.01$) with no significant impact to the overall model performance (Figure H.3). We observe two crossover points where 0-20Y performance becomes worse than the overall model. 1) Between $r = 0.5$ and $r = 0.75$ for FNR ($\Delta_{\text{0-20Y}-\text{All}} = -0.04$ to $0.05$, $p \geq 0.05$). 2) Between $r = 0.75$ and $r = 1$ for AUROC ($\Delta_{\text{0-20Y}-\text{All}} = 0.01$ to $-0.05$, $p = 0.04$).

Let us assume that the change in metric for a group with increasing rate of bias injected is a smooth curve. The intermediate value theorem tells us that the presence of crossover points in certain demographic groups implies that there exists an underdiagnosis rate where the group's model performance is approximately equal to the overall model performance. A more resourced adversary could inject small amounts of adversarial bias at this point to manipulate model performance 1) without being detected, 2) no impact to overall model performance, and 3) no statistically significant inter-group differences. While minor in nature, such attacks could have severe downstream implications on patient health outcomes in the clinical environment.

## Appendix F. Validation with Different Model Architectures

To further validate our findings, we repeat our analysis for age and sex demographic groups in the RSNA dataset using two additional model architectures (ResNet50 and InceptionV3) and compare with our findings with the DenseNet121 model architecture. Figures F.1 and F.2 show the vulnerability and bias selectively for FNR and FOR resp. Table F.1 provides the detailed vulnerability for each demographic group.

We observe that the vulnerability and bias selectivity across FNR and FOR scales well between the three different model architectures. For sex groups, there is inconsistency in the least and most vulnerable groups between the model architectures. For FNR, the female group is the most vulnerable for DenseNet121 but the least vulnerable for ResNet50 and InceptionV3. For FOR, the female group is the most vulnerable for DenseNet121 and ResNet50 but the least vulnerable for InceptionV3. However, for age groups, all three model architectures indicate that the 0-20Y group is the most vulnerable and 40-60Y group is the least vulnerable group across FNR and FOR. We similarly do not include results for the 80+Y group due to small sample sizes.

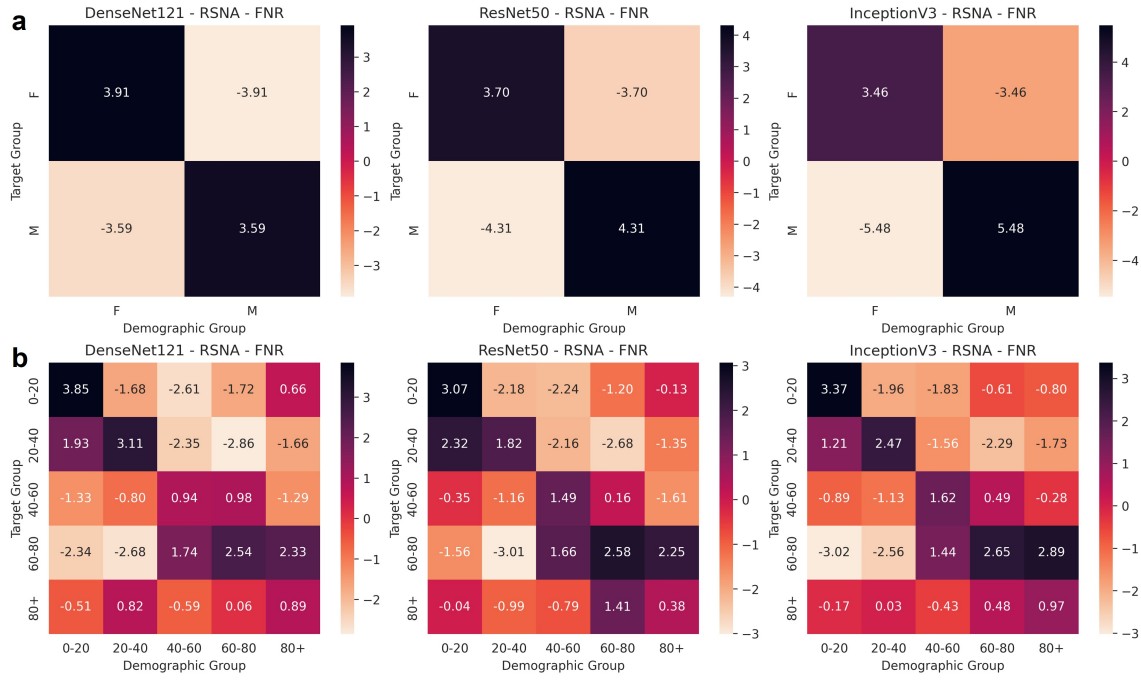

Figure F.1: Vulnerability of FNR for **(a)** sex and **(b)** age groups across DenseNet121 (column 1), ResNet50 (column 2), and InceptionV3 (column 3) model architectures.

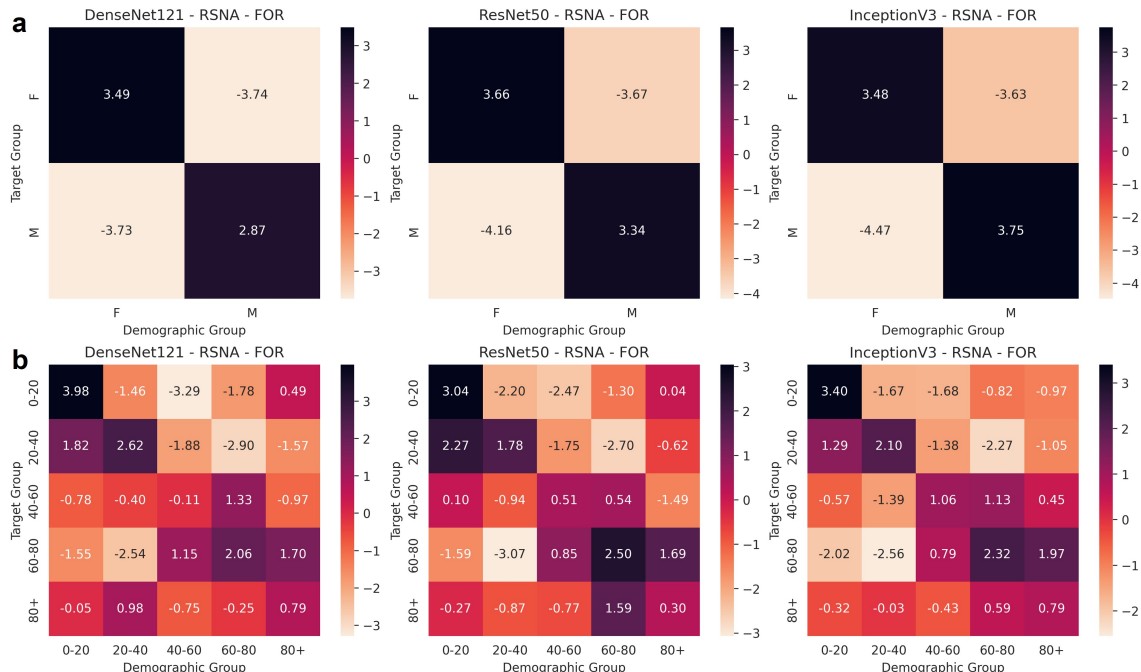

Figure F.2: Vulnerability of FOR for **(a)** sex and **(b)** age groups across DenseNet121 (column 1), ResNet50 (column 2), and InceptionV3 (column 3) model architectures.

Table F.1: Vulnerability $\nu$ values for age and sex demographic groups in the RSNA dataset for DenseNet121 (default), ResNet50, and InceptionV3 DL model architectures across FNR and FOR. For each category, the most vulnerable group (bold) and the least vulnerable group (underline) are indicated.

| Group | FNR | | | FOR | | |
|---|---|---|---|---|---|---|
| | **DenseNet121** | **ResNet50** | **InceptionV3** | **DenseNet121** | **ResNet50** | **InceptionV3** |
| Sex | | | | | | |
| Male | 3.59 | **4.31** | **5.48** | 2.87 | 3.34 | **3.75** |
| Female | **3.91** | 3.70 | 3.46 | **3.49** | **3.66** | 3.48 |
| Age | | | | | | |
| 0-20 | **3.85** | **3.07** | **3.37** | **3.98** | **3.04** | **3.40** |
| 20-40 | 3.11 | 1.82 | 2.47 | 2.62 | 1.78 | 2.10 |
| 40-60 | 0.94 | 1.49 | 1.62 | −0.11 | 0.51 | 1.06 |
| 60-80 | 2.54 | 2.58 | 2.65 | 2.06 | 2.50 | 2.32 |
| 80+ | 0.89[1] | 0.38[1] | 0.97[1] | 0.79 | 0.30[1] | 0.79[1] |

[1] Conclusions lack statistical power due to small sample sizes

## Appendix G. Alternative Vulnerability Measures

For our analysis, we developed three potential different metrics to characterize the vulnerability of a demographic group for adversarial bias attacks. Spearman's rank correlation $r_s$ is used to measure correlation of group vulnerability with sample size.

**Metric 1:** We define this vulnerability metric as the absolute difference in the rate parameters of the logistic regression for the rate of change of metric for the targeted group and the overall model with increasing rate of bias injected. We observe a strong monotonic decreasing relationship between a group's vulnerability and its sample size in the training data across FNR ($r_s = -0.87, p < 0.001$) and FOR ($r_s = -0.74, p < 0.001$) (Figure G.1). However, this metric fails to characterizes a group's vulnerability to bias injection and the impact of label poisoning by identifying all the groups that would be affected when a particular demographic group is targeted.

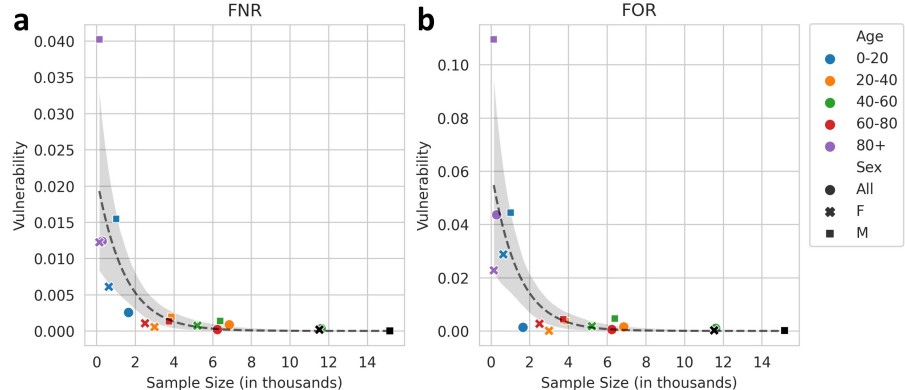

Figure G.1: Correlation of a demographic group's vulnerability with its sample size in the model's training data across (a) FNR and (b) FOR. The logistic regression curve is also plotted.

**Metric 2:** We define this vulnerability metric as the rate parameter of the logistic regression for the rate of change of difference in metric for the targeted group and the overall model with increasing rate of bias injected. We observe a strong monotonic decreasing relationship between a group's vulnerability and its sample size in the training data across FNR ($r_s = -0.89, p < 0.001$) and FOR ($r_s = -0.91, p < 0.001$) (Figure G.2). However, this metric fails to characterizes a group's vulnerability to bias injection and the impact of label poisoning by identifying all the groups that would be affected when a particular demographic group is targeted.

**Metric 3:** We define this vulnerability metric as the rate parameter $\beta$ of the logistic regression for the difference in metric for the targeted group and the overall model with increasing rate of bias injected. We observe a moderate-to-weak monotonic increasing relationship between a group's vulnerability and its sample size in the training data across FNR ($r_s = 0.34, p = 0.18$) and FOR ($r_s = 0.39, p = 0.13$) (Figure G.3). However, this metric accurately characterizes a group's vulnerability to bias injection and the impact of label poisoning by identifying all the groups that would be affected when a particular de-

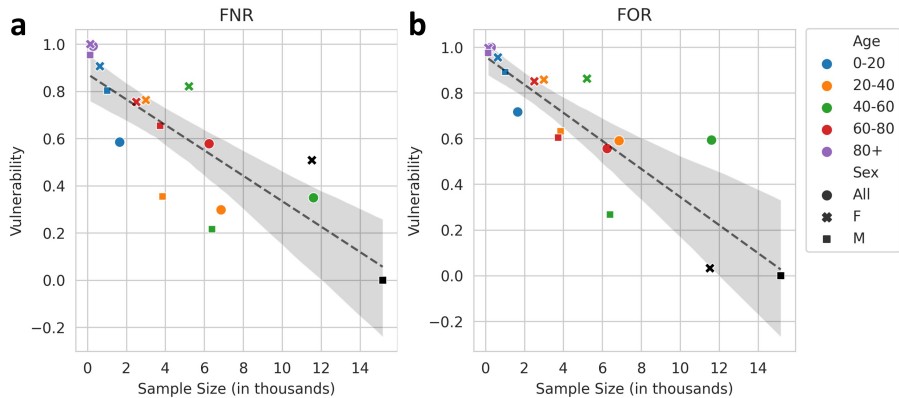

Figure G.2: Correlation of a demographic group's vulnerability with its sample size in the model's training data across (a) FNR and (b) FOR. The logistic regression curve is also plotted.

mographic group is targeted. Therefore, we used this is the metric in our analysis in the main text.

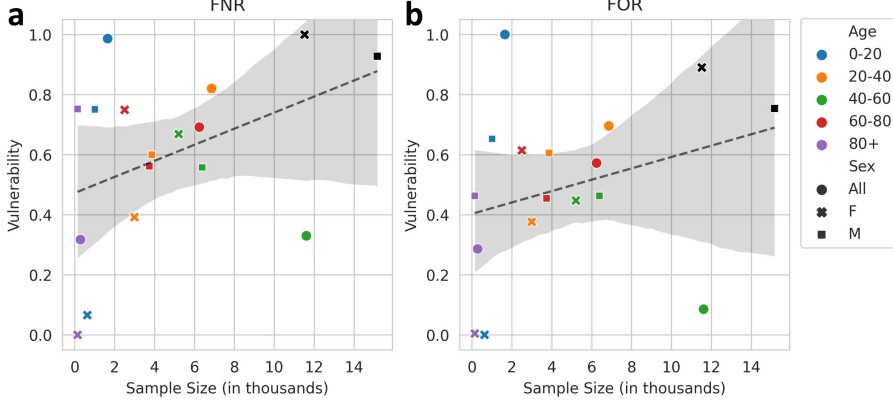

Figure G.3: Correlation of a demographic group's vulnerability with its sample size in the model's training data across (a) FNR and (b) FOR. The logistic regression curve is also plotted.

# Appendix H.  Detailed Figures for Adversarial Bias Attacks

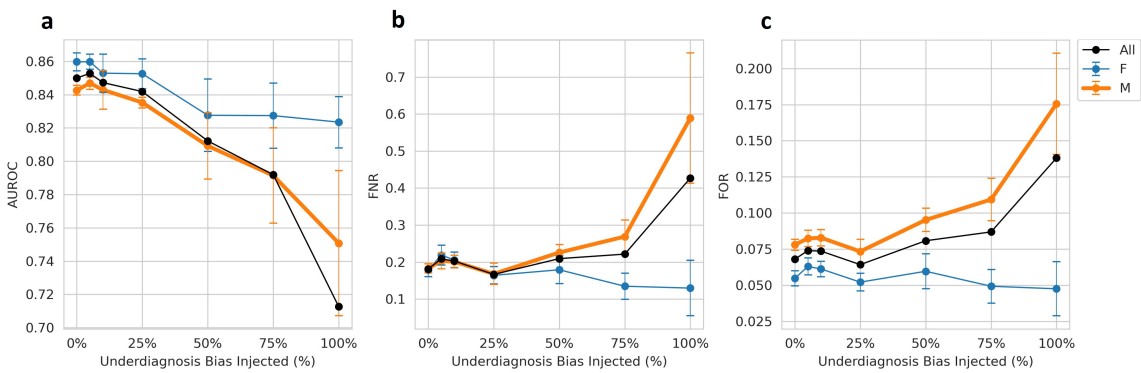

Figure H.1:  Impact of adversarial bias attack on male patients (bold) across **(a)** AUROC, **(b)** FNR, and **(c)** FOR. Metrics provided with mean and 95% CI.

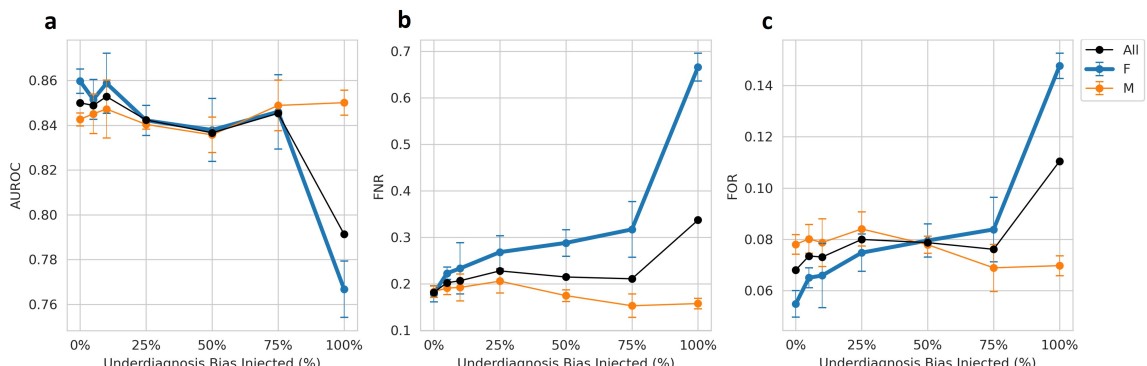

Figure H.2:  Impact of adversarial bias attack on female patients (bold) across **(a)** AUROC, **(b)** FNR, and **(c)** FOR. Metrics provided with mean and 95% CI.

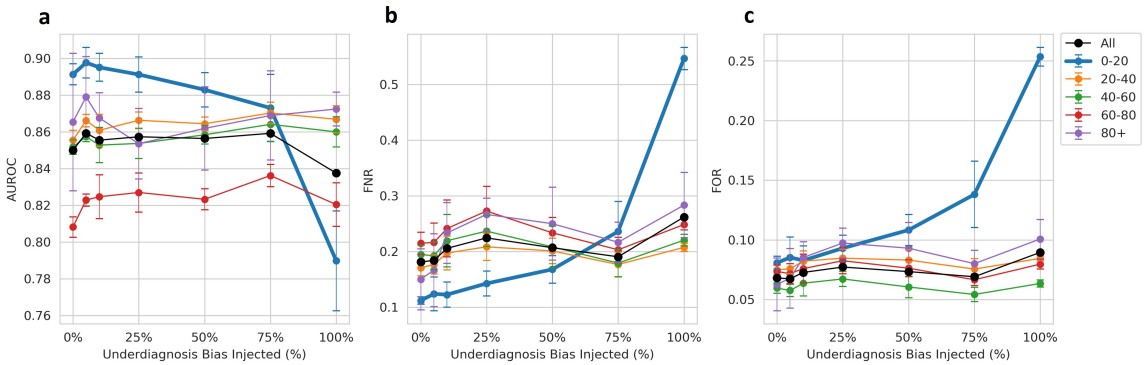

Figure H.3: Impact of adversarial bias attack on 0-20 year old patients (bold) across **(a)** AUROC, **(b)** FNR, and **(c)** FOR. Metrics provided with mean and 95% CI.

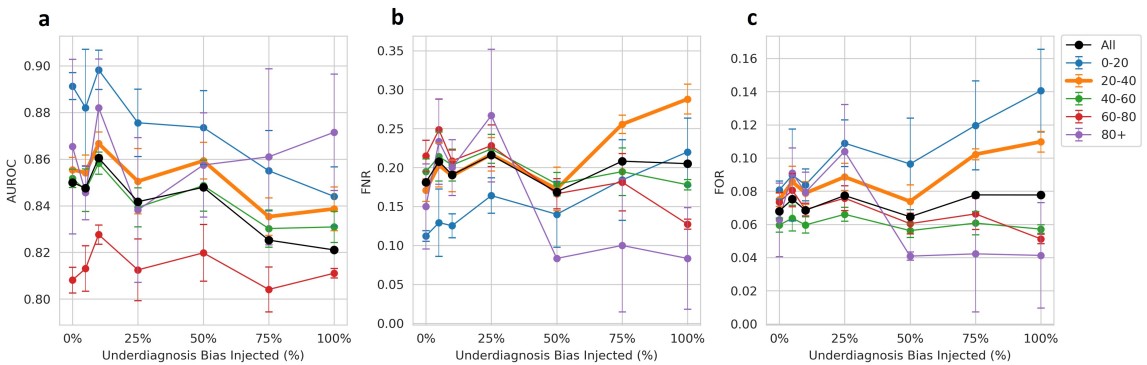

Figure H.4: Impact of adversarial bias attack on 20-40 year old patients (bold) across **(a)** AUROC, **(b)** FNR, and **(c)** FOR. Metrics provided with mean and 95% CI.

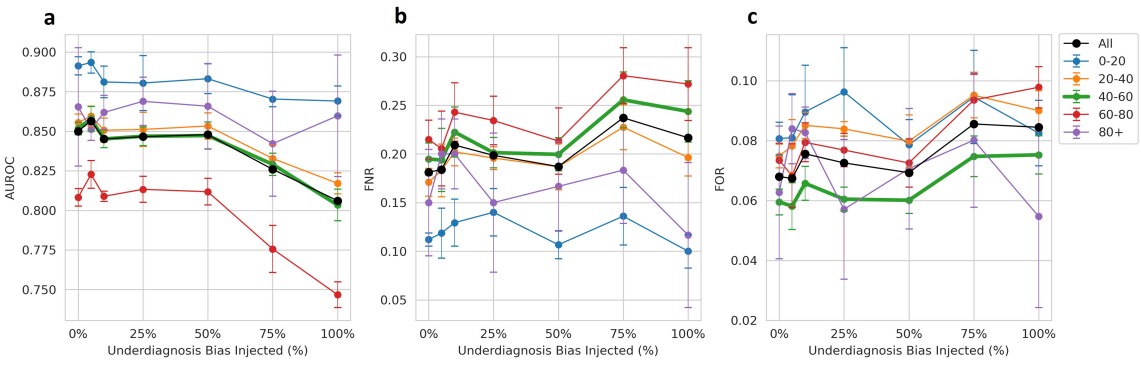

Figure H.5: Impact of adversarial bias attack on 40-60 year old patients (bold) across **(a)** AUROC, **(b)** FNR, and **(c)** FOR. Metrics provided with mean and 95% CI.

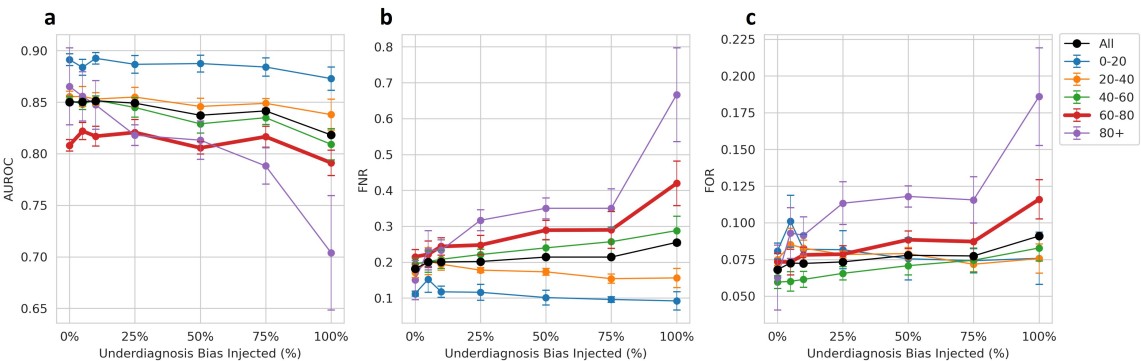

Figure H.6: Impact of adversarial bias attack on 60-80 year old patients (bold) across **(a)** AUROC, **(b)** FNR, and **(c)** FOR. Metrics provided with mean and 95% CI.

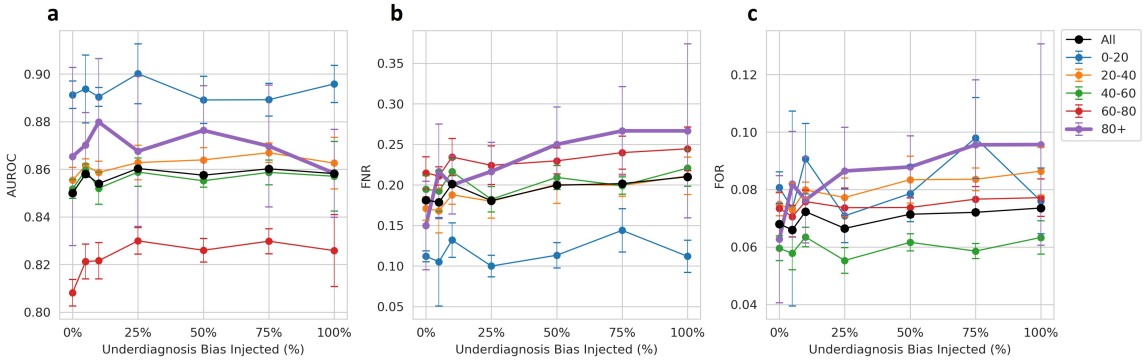

Figure H.7: Impact of adversarial bias attack on 80+ year old patients (bold) across **(a)** AUROC, **(b)** FNR, and **(c)** FOR. Metrics provided with mean and 95% CI.

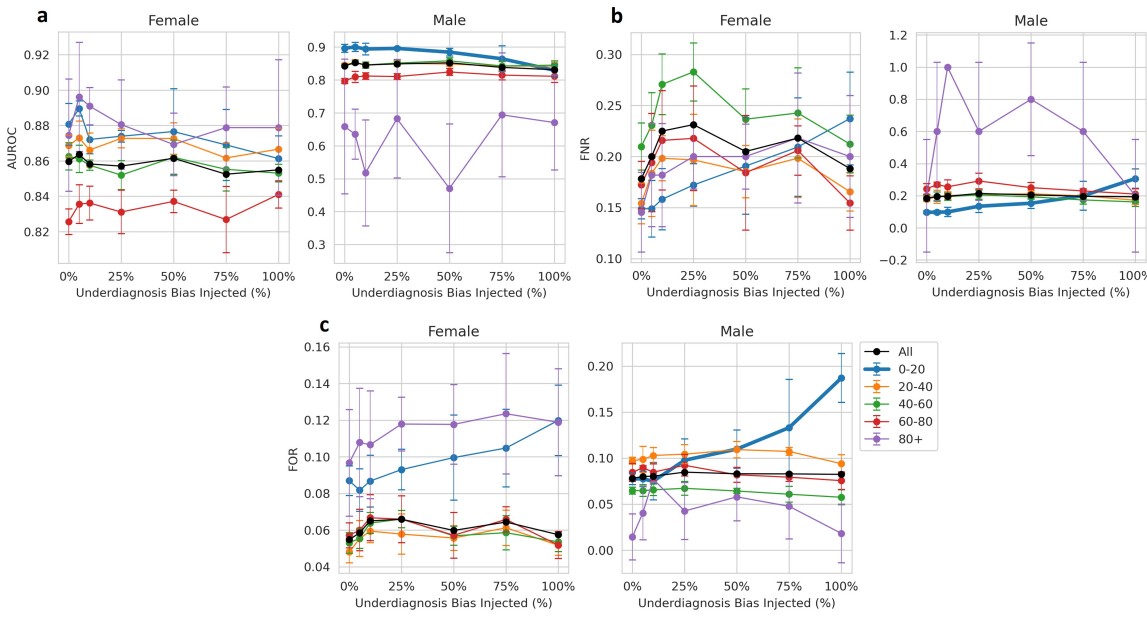

Figure H.8: Impact of adversarial bias attack on 0-20 year old male patients (bold) across **(a)** AUROC, **(b)** FNR, and **(c)** FOR. Metrics provided with mean and 95% CI.

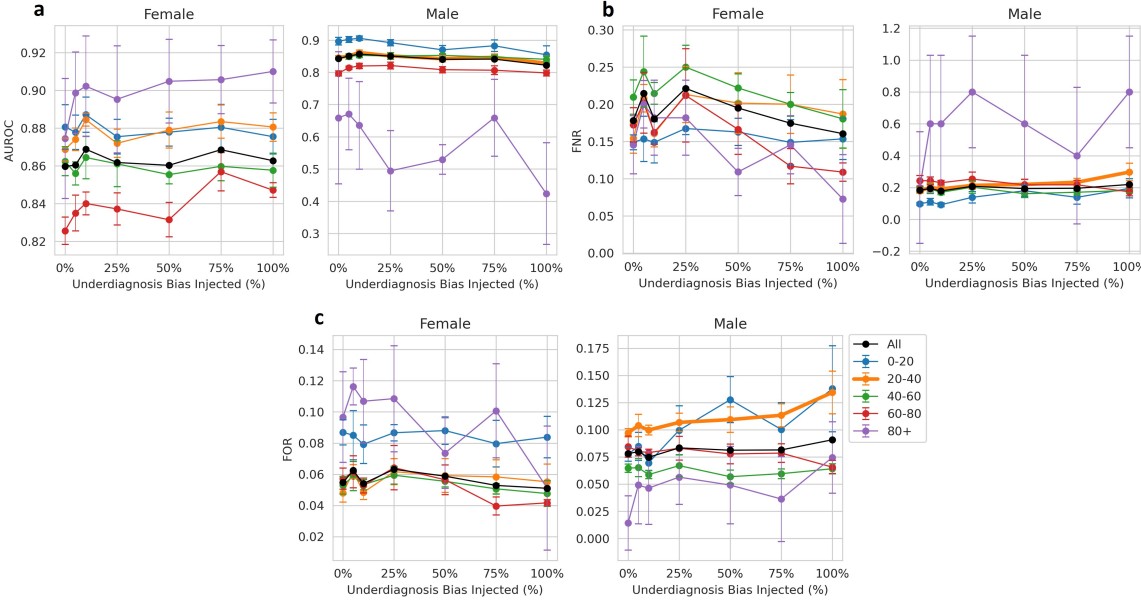

Figure H.9: Impact of adversarial bias attack on 20-40 year old male patients (bold) across **(a)** AUROC, **(b)** FNR, and **(c)** FOR. Metrics provided with mean and 95% CI.

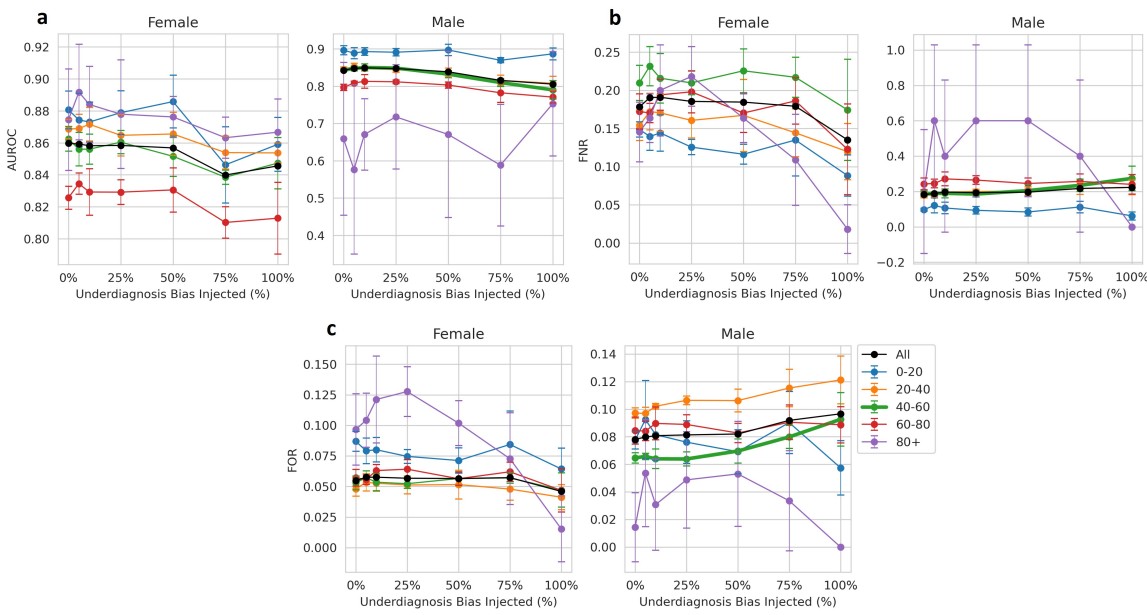

Figure H.10: Impact of adversarial bias attack on 40-60 year old male patients (bold) across **(a)** AUROC, **(b)** FNR, and **(c)** FOR. Metrics provided with mean and 95% CI.

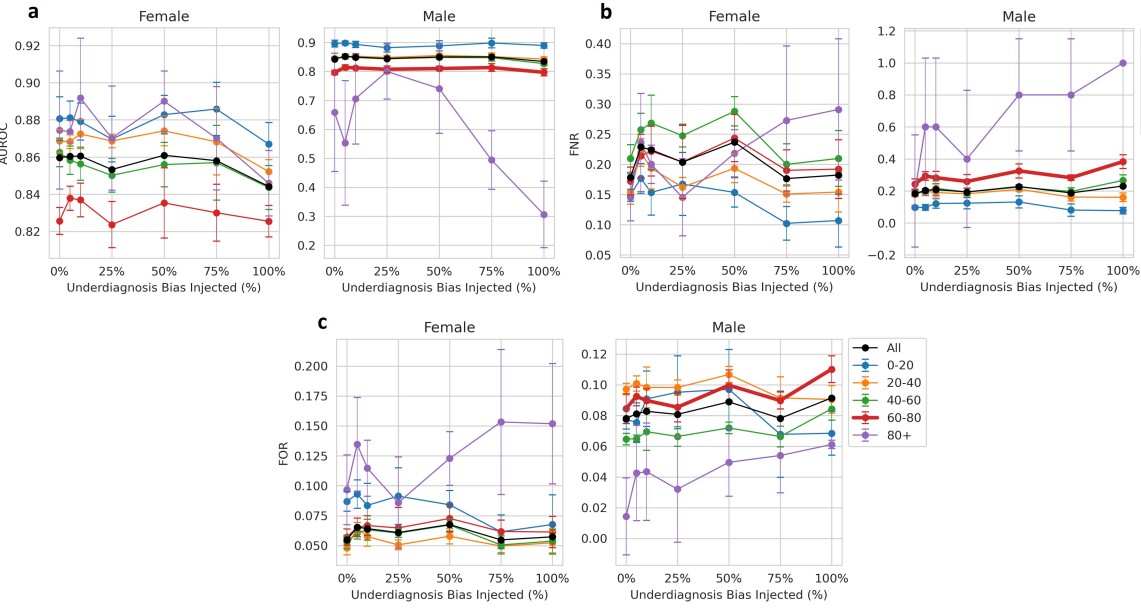

Figure H.11: Impact of adversarial bias attack on 60-80 year old male patients (bold) across **(a)** AUROC, **(b)** FNR, and **(c)** FOR. Metrics provided with mean and 95% CI.

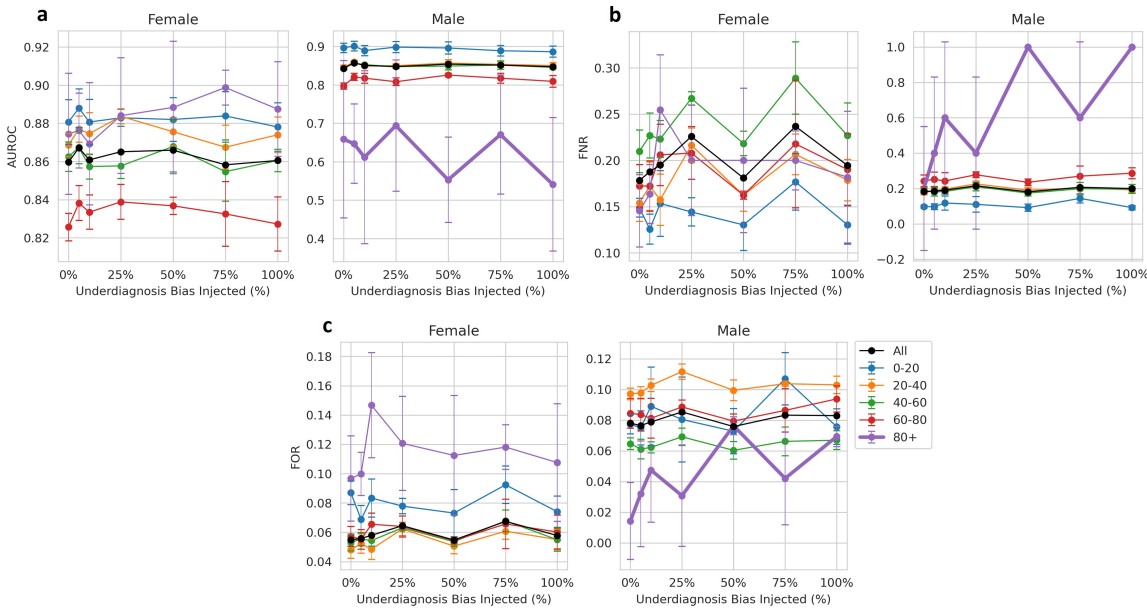

Figure H.12:  Impact of adversarial bias attack on 80+ year old male patients (bold) across **(a)** AUROC, **(b)** FNR, and **(c)** FOR. Metrics provided with mean and 95% CI.

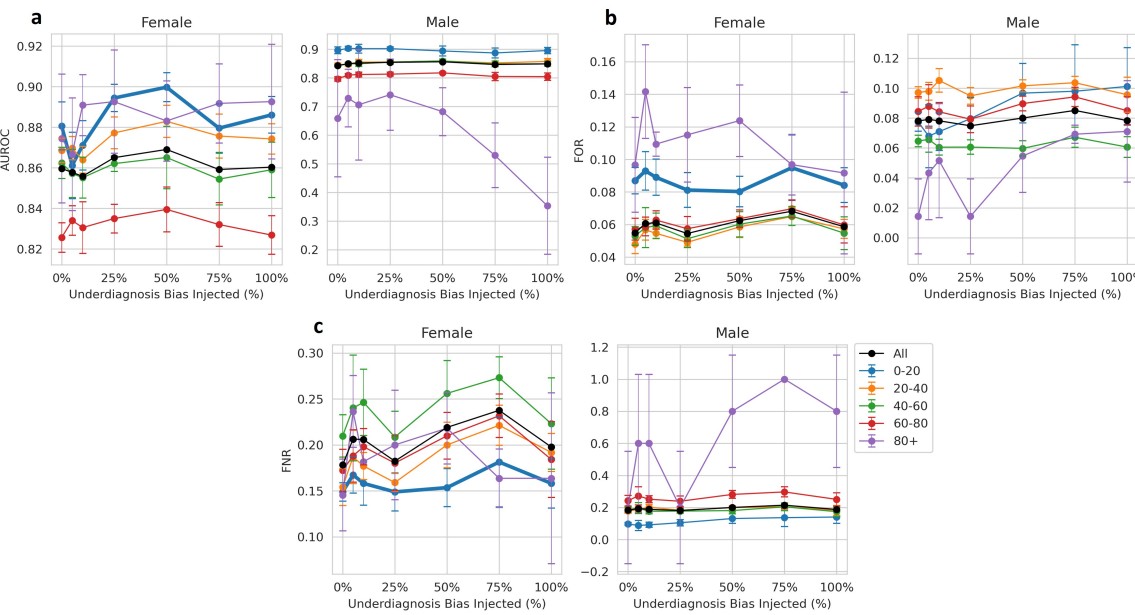

Figure H.13:  Impact of adversarial bias attack on 0-20 year old female patients (bold) across **(a)** AUROC, **(b)** FNR, and **(c)** FOR. Metrics provided with mean and 95% CI.

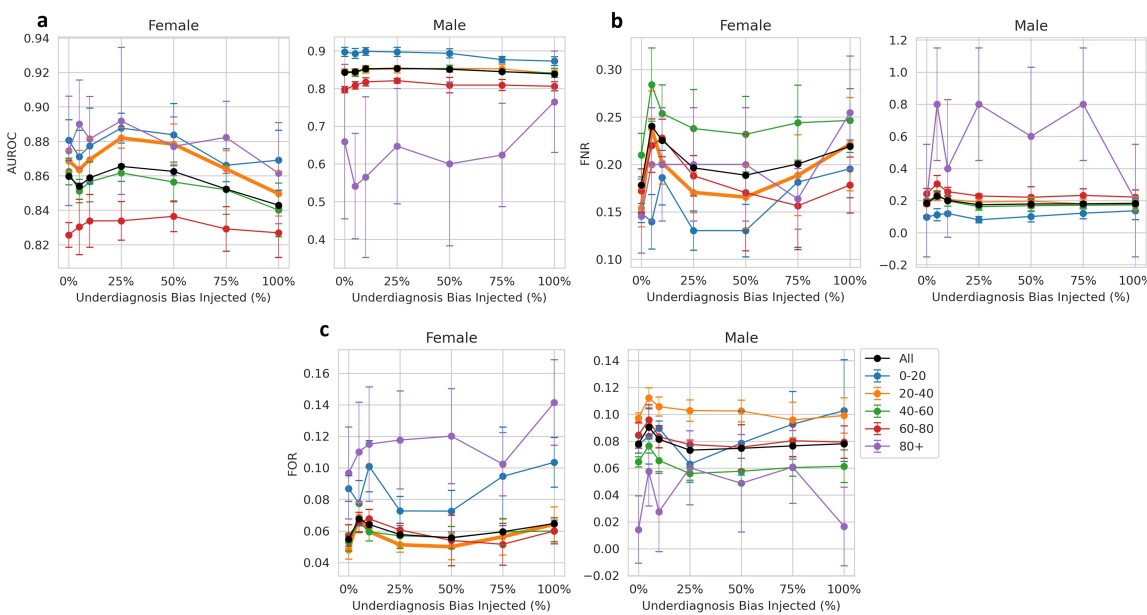

Figure H.14: Impact of adversarial bias attack on 20-40 year old female patients (bold) across **(a)** AUROC, **(b)** FNR, and **(c)** FOR. Metrics provided with mean and 95% CI.

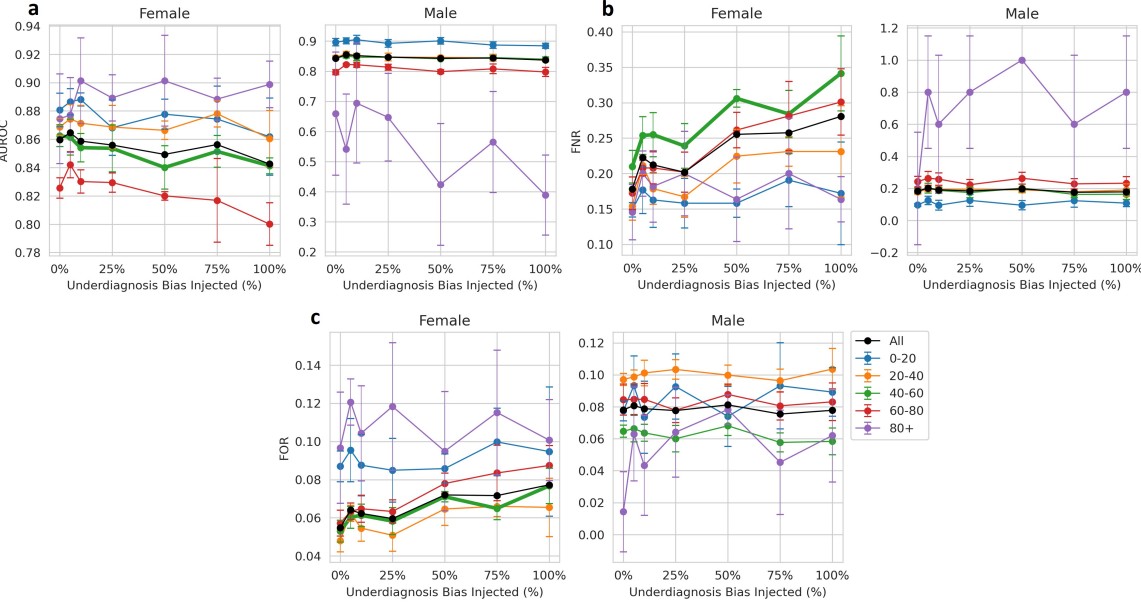

Figure H.15: Impact of adversarial bias attack on 40-60 year old female patients (bold) across **(a)** AUROC, **(b)** FNR, and **(c)** FOR. Metrics provided with mean and 95% CI.

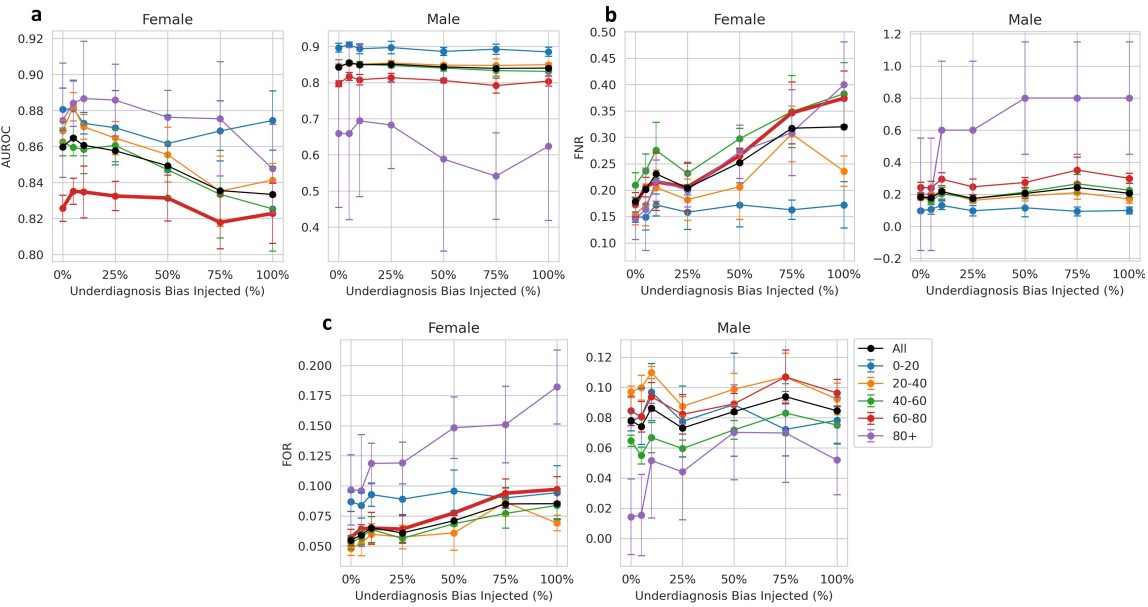

Figure H.16: Impact of adversarial bias attack on 60-80 year old female patients (bold) across **(a)** AUROC, **(b)** FNR, and **(c)** FOR. Metrics provided with mean and 95% CI.

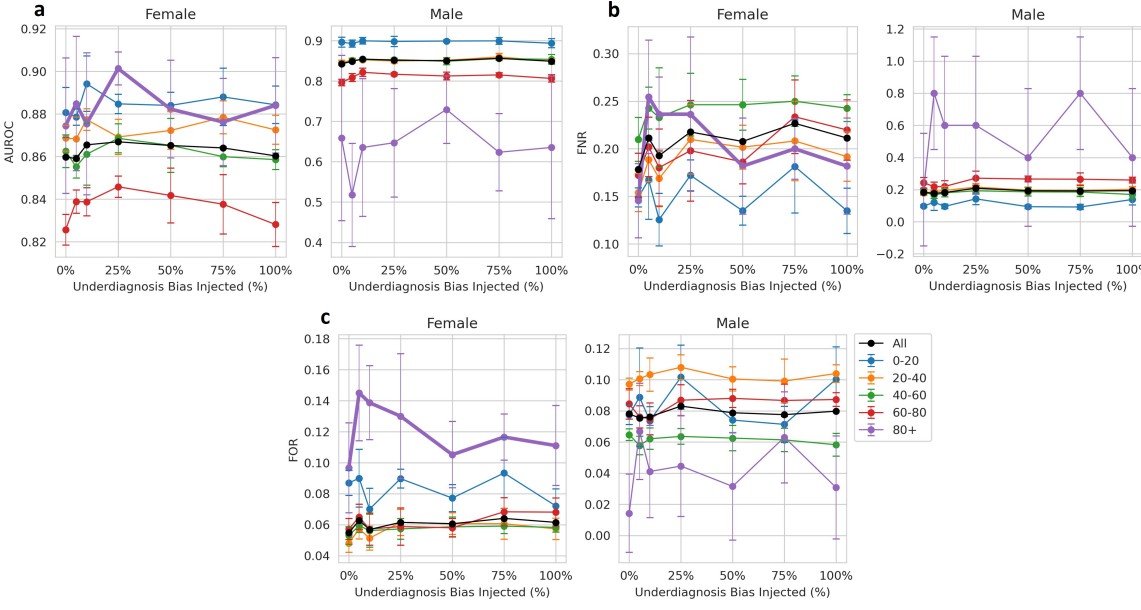

Figure H.17: Impact of adversarial bias attack on 80+ year old female patients (bold) across **(a)** AUROC, **(b)** FNR, and **(c)** FOR. Metrics provided with mean and 95% CI.

