# OpenReview forum: "Hidden in Plain Sight: Undetectable Adversarial Bias Attacks on Vulnerable Patient Populations"
_MIDL.io/2024/Conference — MIDL 2024 Oral_

### Official Review · Reviewer_NvyJ · 2024-02-27

**Confidence:** 4
**Preliminary Rating:** 3
**Final Rating:** 4

**Summary:**

A demographically targeted label poisoning attack has been a cause of the bias from deep learning models. In this work, the authors quantified and demonstrated the degraded performance of minority groups, and proposed a method without impacting overall model performance.

**Strengths:**

- The authors have shown that adversarial attacks have a great impact other than just noise.
- A minority’s vulnerability was introduced to undetectable adversarial bias attacks.
- The authors showed the vulnerability was connected to the sample size in the training set.

**Weaknesses:**

- The authors only performed the experiments on a single dataset. Results on more datasets are strongly encouraged to provide stronger evidence of the proposal.
- The authors randomly sampled the dataset into training/validation and testing sets with 5-folds. However in Appendix B, there’s no standard deviation used. The authors are encouraged to provide the information regarded.
- Instead of showing the plots, the authors could use a table for better demonstration of the proposed results.
- The authors only performed results on DenseNet121, which could be an indicator that the vulnerability is with the DenseNet121 with respect to Chest X-Ray datasets. More models could be included.
- Could the authors include more vulnerability metrics, other than the proposed one, to show that the proposed is actually working?

**Detailed Comments:**

- The authors only performed the experiments on a single dataset. Results on more datasets are strongly encouraged to provide stronger evidence of the proposal.
- The authors randomly sampled the dataset into training/validation and testing sets with 5-folds. However in Appendix B, there’s no standard deviation used. The authors are encouraged to provide the information regarded.
- Instead of showing the plots, the authors could use a table for better demonstration of the proposed results.
- The authors only performed results on DenseNet121, which could be an indicator that the vulnerability is with the DenseNet121 with respect to Chest X-Ray datasets. More models could be included.
- Could the authors include more vulnerability metrics, other than the proposed one, to show that the proposed is actually working?

**Justification Of Final Rating:**

I would like to express my sincere thanks to the authors for their thorough classification efforts. The inclusion of additional experiments, the expansion of classifications, and the enhancement of illustrations have successfully addressed all of my concerns.

=====Update=====Thus, I would like to vote for acceptance.

**Justification Of The Preliminary Rating:**

- The authors only performed the experiments on a single dataset. Results on more datasets are strongly encouraged to provide stronger evidence of the proposal.
- The authors randomly sampled the dataset into training/validation and testing sets with 5-folds. However in Appendix B, there’s no standard deviation used. The authors are encouraged to provide the information regarded.
- Instead of showing the plots, the authors could use a table for better demonstration of the proposed results.
- The authors only performed results on DenseNet121, which could be an indicator that the vulnerability is with the DenseNet121 with respect to Chest X-Ray datasets. More models could be included.
- Could the authors include more vulnerability metrics, other than the proposed one, to show that the proposed is actually working?

**Questions To Address In The Rebuttal:**

- The authors only performed the experiments on a single dataset. Results on more datasets are strongly encouraged to provide stronger evidence of the proposal.
- The authors randomly sampled the dataset into training/validation and testing sets with 5-folds. However in Appendix B, there’s no standard deviation used. The authors are encouraged to provide the information regarded.
- Instead of showing the plots, the authors could use a table for better demonstration of the proposed results.
- The authors only performed results on DenseNet121, which could be an indicator that the vulnerability is with the DenseNet121 with respect to Chest X-Ray datasets. More models could be included.
- Could the authors include more vulnerability metrics, other than the proposed one, to show that the proposed is actually working?

---

> ### Author Response · Authors · 2024-03-18
> **Rebuttal to Reviewer 3**
>
> Thank you for taking the time to provide valuable insights on our submission. We have addressed all concerns that were raised, and we feel confident that the manuscript has been improved in this process. Please see below the point-by-point response to the comments (in bold):
>
> **1. The authors only performed the experiments on a single dataset. Results on more datasets are strongly encouraged to provide stronger evidence of the proposal.**
>
> Please refer to our general rebuttal response #2 for external validation of our findings.
>
> **2. The authors randomly sampled the dataset into training/validation and testing sets with 5-folds. However in Appendix B, there’s no standard deviation used. The authors are encouraged to provide the information regarded.**
>
> We have revised the submission to include standard deviation for sample sizes for the training and validation splits for 5-fold cross-validation.
>
> **3. Instead of showing the plots, the authors could use a table for better demonstration of the proposed results.**
>
> We have updated the results in our submission with an emphasis on the vulnerability and transferability of bias across demographic groups.
>
> **4. The authors only performed results on DenseNet121, which could be an indicator that the vulnerability is with the DenseNet121 with respect to Chest X-Ray datasets. More models could be included.**
>
> Please refer to our general rebuttal response #3 for results from experiments on additional model architectures.
>
> **5. Could the authors include more vulnerability metrics, other than the proposed one, to show that the proposed is actually working?**
>
> We thank the reviewer for this question. In response to this question, we developed different metrics to characterize vulnerability for comparison and in the process, developed a more appropriate metric. We have included the comparison between the final vulnerability metric used in our analysis and other vulnerability metrics (**Appendix G**). The final vulnerability metric not only characterizes a demographic group’s vulnerability to undetectable adversarial bias injection but also characterizes the impact of label poisoning by identifying all the demographic groups that would be affected when a particular demographic group is targeted.

---

### Official Review · Reviewer_q6Zb · 2024-02-27

**Confidence:** 3
**Preliminary Rating:** 3
**Final Rating:** 4

**Summary:**

This paper investigates how label poisoning (e.g. wrongly/randomly changing the labels in train set) for some specific subgroups can affect the overall performance and performance on the targeted subgroup. They focused only on the attacks on labels of the target subgroups. They found when the attack is targeting on underrepresented subgroup, the overall performance is not significant affected, which makes this attack hard to detect. On the contrast, for majority subgroups, the overall performance is more linked to the degrade of subgroup performance.

**Strengths:**

This paper focus on label poisoning on the targeted subgroup. Specifically, they investigate the effect of attack at a subgroup on the performance of that subgroup and overall performance and they reveal for underrepresented subgroups, the attack is more undetectable as it does not significantly degrade the overall performance.

The method is easy to follow and interesting.

**Weaknesses:**

- They only evaluate on one dataset, which raises concerns about generalization of their findings.
- The motivation is not clear. I would expect that label poisoning on minority subgroup to have less effect on overall performance. But what would be the impact of it? Does that mean we need to be very careful with labels when dealing with minority subgroups as it is hard to detect error for them? I think a better description of the findings is helpful.
- Also, do you have any thoughts on how to mitigate this undetectable error in real-world scenarios?
- Here, majority and minority subgroups are the same for train sets and test sets. But what if for train set, say, male is the majority subgroup, and for test set/ real world data, male is the minority group, what would be the effect?

**Detailed Comments:**

- Figure 1 caption is not clear. Maybe just say you wrongly change the label for a subgroup. I find the current caption is very general and vague.
- Some description are unnecessarily over-complex. The method itself is very easy, mislabelling a ratio of labels for a subgroup.

**Justification Of Final Rating:**

Idea is interesting, but experiments only performed on one dataset. Also, the discovered finding, that if we mislabel a minority subgroup, then its effect on the overall performance would be less than if we mislabel a majority subgroup, seems to be not very informative, as it is as expected. And there is no discussion on how to mitigate this issue.

====update=====
Authors have performed experiments on additional experiments, and talked how to potentially mitigate this issue.

**Justification Of The Preliminary Rating:**

Idea is interesting, but experiments only performed on one dataset. Also, the discovered finding, that if we mislabel a minority subgroup, then its effect on the overall performance would be less than if we mislabel a majority subgroup, seems to be not very informative, as it is as expected. And there is no discussion on how to mitigate this issue.

**Questions To Address In The Rebuttal:**

Same as “weakness” but I attach here:

- The motivation is not clear. I would expect that label poisoning on minority subgroup to have less effect on overall performance. But what would be the impact of it? Does that mean we need to be very careful with labels when dealing with minority subgroups as it is hard to detect error for them? I think a better description of the findings is helpful.
- Also, do you have any thoughts on how to mitigate this undetectable error in real-world scenarios?
- Here, majority and minority subgroups are the same for train sets and test sets. But what if for train set, say, male is the majority subgroup, and for test set/ real world data, male is the minority group, what would be the effect?

**Special Issue:**

No

---

> ### Author Response · Authors · 2024-03-18
> **Rebuttal to Reviewer 2**
>
> Thank you for taking the time to provide valuable insights on our submission. We have addressed all concerns that were raised, and we feel confident that the manuscript has been improved in this process. Please see below the point-by-point response to the comments (in bold):
>
> **1. They only evaluate on one dataset, which raises concerns about generalization of their findings.**
>
> Please refer to our general rebuttal response #2 for external validation of our findings.
>
> **2. The motivation is not clear. I would expect that label poisoning on minority subgroup to have less effect on overall performance. But what would be the impact of it? Does that mean we need to be very careful with labels when dealing with minority subgroups as it is hard to detect errors for them? I think a better description of the findings is helpful.**
>
> The motivation for this study was two-fold.
> - First, develop bias injection attacks that could selectively attack a specific demographic group without affecting the performance on other groups. Furthermore, evaluate if this selective bias can propagate to an external test set. Our results indicate that not only is it possible to selectively target specific demographic groups but also that this selective bias injection does propagate to external test datasets.
> - Second, evaluate when such attacks could potentially go undetected and develop a quantifiable metric to characterize such scenarios. Our results indicate that it is possible to create undetectable data poisoning attacks. We further developed a vulnerability metric, $\nu$ to characterize the undetectability of the targeted demographic groups as well as the impact of bias injection on the remaining non-targeted demographic populations.
>
> For the use-case that the reviewer pointed out, label poisoning on a minority group with a lesser effect on the overall performance may go undetected during model training. Consider two AI vendors that developed AI models with the same overall performance, except that one AI vendor’s model contains this “hidden bias”. The implication of this “hidden bias” is that if a hospital system ends up with the model with “hidden bias” this bias would propagate to real-world model deployment and bias the clinical workflow impacting lives of patients within the biased demographic group.
>
> We also would like to point the reviewer to our results that demonstrate the possibility of creating undetectable bias attacks on well represented population groups. For example, the complete female subgroup represents 44% of the training set but is still vulnerable to undetectable bias attacks. At 75% data poisoning, the female population suffers a significant degradation in the performance without any significant impact to the overall performance across evaluation metrics.
>
> Therefore, we need to ensure we take appropriate measures to mitigate undetectable bias injection attacks on all vulnerable populations, both during data curation and model training. We have updated our description of findings to make this clearer.
>
> **3. Also, do you have any thoughts on how to mitigate this undetectable error in real-world scenarios?**
>
> Please refer to our general rebuttal response #1 for potential defenses against adversarial bias attacks.
>
> **4. Here, majority and minority subgroups are the same for train sets and test sets. But what if for train set, say, male is the majority subgroup, and for test set/ real world data, male is the minority group, what would be the effect?**
>
> The reviewer raises an important point about diversity in the relative representation of different sub-groups in the training and test sets. We experimentally evaluated our method on external datasets in general rebuttal response #2. Our study evaluated two aspects of bias injection – transferability and vulnerability. The following points discusses both these aspects in such train-test scenarios:
> - Transferability: Our results demonstrated that the injected bias was successfully propagated to the test set across all demographic groups irrespective of their relative representation in the training set.
> - Vulnerability: The vulnerability metric was defined specifically for training setup to evaluate if a “hidden bias” can be injected in a demographic group during the training process. In real-world deployment, the ground truth for the test cases may not be available for vulnerability analysis.
>
> When ground truth is available for the test sets, detectability of  transferred bias from the training set can be evaluated by computing vulnerability values for every group in the test set. We have updated our results with vulnerability values across both training and test sets (**Table A.1** and **Figure 4**). **Table A.1** shows 40-60Y is the majority group in the training set while the 60-80Y is the majority group in the test sets (CheXpert, MIMIC). Despite being the majority group in the test sets, the 60-80Y group demonstrated the maximum FNR vulnerability.

---

### Official Review · Reviewer_3qA9 · 2024-02-28

**Confidence:** 4
**Preliminary Rating:** 4
**Recommendation:** Oral
**Final Rating:** 5

**Summary:**

The authors explore an although lesser-known, but very important problem, which are the effect of adversarial bias attacks, impacting the performance of their trained models on the most vulnerable patient groups, without making a noticable impact on the overall model performance. The authors show a direct correlation between the representation of certain patient groups and their vulnerability towards such attacks.

**Strengths:**

The authors highlight the importance of adversarial label poisoning, and more importantly focus on the vulnerability of different patient groups.
The experiments are well-designed and the statistical analysis is insightful.

**Weaknesses:**

The authors thoroughly explore the issue at hand, and suggest that for better transparency, datasets should report sub-group analysis, however I am missing any suggestions with regards to solving the problem when training DL models. Should we focus on data augmentation in under-represented groups? Are there methods to make the model more robust towards such attacks?

**Detailed Comments:**

The online repository could improve by including a list of the required package dependencies.

**Justification Of Final Rating:**

I would like to thank the authors for thoroughly addressing all my concerns, and the concerns of all the other authors. Based on the numerous adjustments and improvements made to the manuscript, I am updating my final rating.

**Justification Of The Preliminary Rating:**

The paper provides an interesting read, the experiments are well-designed and thorough. However the reviewer is left to wonder "What now?" Given the poorly-described public datasets with regards to sub-group analysis, the paper lacks the logical next step as for how should the presented results change how researchers develop their models?

**Questions To Address In The Rebuttal:**

Now that the authors brought awareness to patient group vulnerability, do they have any suggestions on how to prepare a model for such attacks? Should it be avoided when preparing the data, such as over-sampling under-represented groups? Or should it be avoided during model training perhaps?
The described adversarial bias attacks assume a detailed knowledge of the training dataset. Without disclosing any sub-group analysis of the dataset, is it possible to target under-represented populations? As the authors acknowledged a limitation of their work is that they assumed bias injection in only one demographic group at a time. How would someone find this group without access to a well-described training dataset?

**Special Issue:**

No

---

> ### Author Response · Authors · 2024-03-18
> **Rebuttal to Reviewer 1**
>
> Thank you for taking the time to provide valuable insights on our submission. We have addressed all concerns that were raised, and we feel confident that the manuscript has been improved in this process. Please see below the point-by-point response to the comments (in bold):
>
> **1. The online repository could improve by including a list of the required package dependencies.**
>
> We have updated the online repository with appropriate documentation and the list of required package dependencies.
>
> **2. Now that the authors brought awareness to patient group vulnerability, do they have any suggestions on how to prepare a model for such attacks?**
>
> Please refer to our general rebuttal response #1 for potential defenses against adversarial bias attacks.
>
> **3. Should it be avoided when preparing the data, such as over-sampling under-represented groups? Or should it be avoided during model training perhaps?**
>
> Adversarial bias attacks could occur in both scenarios and appropriate defenses would need to be implemented to avoid undetectable bias injection in both cases. Over-sampling under-represented groups can potentially make bias more detectable during validation using fairness metrics like equal opportunity and equalized odds. The general rebuttal response #1 details different avenues for bias attacks with potential defenses.
>
> **4. The described adversarial bias attacks assume a detailed knowledge of the training dataset. Without disclosing any sub-group analysis of the dataset, is it possible to target under-represented populations?**
>
> Without access to dataset demographics in the training data, it is still possible to identify and target underrepresented groups.
>
> _During Data Curation_
>
> During data curation, bias can be introduced in DL models (accidentally or intentionally) via errors in automated NLP labelers and clinical biases [1,2]. Prior literature has shown that not only can these automated labelers introduce errors [1], but also can be biased towards underrepresented groups [3]. Furthermore, clinical biases (human biases, financial conflicts of interest, etc.) may introduce bias in the training dataset [1,4].  Prior to model training, an adversary could introduce bias in underrepresented groups using a man-in-the-middle or backdoor attack without requiring access to dataset demographics.
>
> _During Training_
>
> Prior literature has indicated that DL models can predict demographic information with high accuracy using just the images and could be used as a potential method of identifying vulnerable groups in the training dataset [5,6]. Furthermore, lack of demographic information would in-fact hamper identification of undetectable biases injected during the training process.
>
> **5. As the authors acknowledged, a limitation of their work is that they assumed bias injection in only one demographic group at a time. How would someone find this group without access to a well-described training dataset?**
>
> Please refer to our response to comment #4 above for how an adversary can target an underrepresented group without access to a dataset demographics.
>
> **References:**
>
> [1] Cohen, J., Hashir, M., Brooks, R., and Bertrand, H. On the limits of cross-domain generalization in automated X-ray prediction. In Medical Imaging with Deep Learning, pp. 136-155. PMLR, 2020.
>
> [2] Seyyed-Kalantari, L., Zhang, H., McDermott, M., Chen, I., and Ghassemi, M. Underdiagnosis bias of artificial intelligence algorithms applied to chest radiographs in under-served patient populations. Nature Medicine, 27(12):2176-2182, 2021.
>
> [3] Zhang, H., Lu, A., Abdalla, M., McDermott, M., and Ghassemi, M. Hurtful words: quantifying biases in clinical contextual word embeddings. In Proceedings of the ACM Conference on Health, Inference, and Learning, pp. 110-120. 2020.
>
> [4] Friedman, L., De, S., Almberg, K., and Cohen, R. Association between financial conflicts of interest and International Labor Office classifications for black lung disease. Annals of the American Thoracic Society, 18(10):1634-1641, 2021.
>
> [5] Yi, P., Wei, J., Kim, T., Shin, J., Sair, H., Hui, F., Hager, G., and Lin, C. Radiology “forensics”: determination of age and sex from chest radiographs using deep learning. Emergency Radiology 28:949-954, 2021.
>
> [6] Gichoya, J., Banerjee, I., Bhimireddy, A., Burns, J., Celi, L., Chen, L., Correa, R., et al. AI recognition of patient race in medical imaging: a modelling study. The Lancet Digital Health 4(6):e406-e414, 2022.

---

### Author Response · Authors · 2024-03-18
**General Rebuttal**

We thank the reviewers for their valuable feedback. We have updated the submission with this feedback. In this general rebuttal, we aim to address the common concerns raised by the reviewers and provide further validation of our findings.

**1. Defenses**

Adversarial bias attacks can potentially occur both during data curation and training. We identify the following potential avenues for undetectable bias attacks and talk about potential defenses:

_During Data Curation_

During data curation, bias can be introduced in DL models (accidentally or intentionally) via errors in automated NLP labelers and clinical biases [1,2]. Prior literature has shown that not only can these automated labelers introduce errors [1], but also can be biased towards underrepresented groups [3]. Furthermore, clinical biases (human biases, etc.) may introduce bias in the training dataset [1].  Prior to model training, an adversary could introduce bias in underrepresented groups using a man-in-the-middle or backdoor attack without requiring access to dataset demographics. As a potential defense mechanism, the annotation pipeline needs to be extensively evaluated for bias against all demographic groups to avoid any undetectable bias injection into the trained models.

_During Model Training_

The proposed vulnerability metric, $\nu$ characterizes a subgroup’s vulnerability to undetectable bias attacks. We recommend doing a vulnerability analysis to identify potentially vulnerable groups in the training set whose performance can be closely monitored during training to avoid undetectable bias attacks.

The loss functions implemented during model training only use overall model performance on the validation dataset to guide the training process. As a result, bias attacks shown in this work may get encoded without being detected as they do not change the overall performance of the model. As a potential defense, monitoring the model’s performance across all demographic groups would reveal any bias injection attacks on a particular demographic group during the training process. However, as mentioned in the manuscript, this reinforces the importance of including demographic information in the medical imaging datasets.

Since bias attacks using label poisoning are underexplored in literature and are capable of avoiding many pre-existing defense techniques based on knowledge distillation [4, 5], we believe that exploring the potential for such defenses warrants future work.

**2. External Validation**

We used the CheXpert and MIMIC datasets as external test sets to evaluate whether adversarial bias attacks result in biased DL models. (**Section 3.2** and **Figure 4**). We observe that bias attacks exhibited high-selectivity for bias in the targeted group as indicated by a positive $\nu$ and this finding scales to the external datasets. For example, when targeting the 20-40Y group, both 0-20Y and 20-40Y group were affected from the internal RSNA ($\nu=1.93$ vs $3.11$) to CheXpert ($\nu=2.03$ vs $2.03$) and MIMIC ($\nu=1.51$ vs $1.48$).

**3. Experiments on Different Model Architectures**

We repeat our analysis for age and sex demographic groups in the RSNA dataset using two additional model architectures (ResNet50 and InceptionV3) and compare with our findings with the DenseNet121 model architecture (**Appendix F**)

We observe that the vulnerability and bias selectivity across FNR and FOR scales well between the three architectures. For sex groups, there is inconsistency in the least and most vulnerable groups between the model architectures. For FNR, the female group is the most vulnerable for DenseNet121 but the least vulnerable for ResNet50 and InceptionV3. For FOR, the female group is the most vulnerable for DenseNet121 and ResNet50 but the least vulnerable for InceptionV3. For age groups, all three architectures indicate that 0-20Y group is the most vulnerable and 40-60Y group is the least vulnerable group across FNR and FOR.

**References:**

[1] Cohen, J., Hashir, M., Brooks, R., and Bertrand, H. On the limits of cross-domain generalization in automated X-ray prediction. In Medical Imaging with Deep Learning, pp. 136-155. PMLR, 2020.

[2] Seyyed-Kalantari, L., Zhang, H., McDermott, M., Chen, I., and Ghassemi, M. Underdiagnosis bias of artificial intelligence algorithms applied to chest radiographs in under-served patient populations. Nature Medicine, 27(12):2176-2182, 2021.

[3] Zhang, H., Lu, A., Abdalla, M., McDermott, M., and Ghassemi, M. Hurtful words: quantifying biases in clinical contextual word embeddings. In Proceedings of the ACM Conference on Health, Inference, and Learning, pp. 110-120. 2020.

[4] Jha, R., Hayase, J., Oh, S. Label poisoning is all you need. Advances in Neural Information Processing Systems, 36, 2024.

[5] Hayase, J., Kong, W., Somani, R., Oh, S. Spectre: Defending against backdoor attacks using robust statistics. In International Conference on Machine Learning, pp. 4129-4139, PMLR, 2021.

---

### Comment · Area_Chair_Exnb · 2024-03-18
**Invitation to respond to authors**

Dear reviewers,

The authors have prepared responses to your comments, which you should now be able to see in OpenReview. We encourage you to reply to their comments, and where necessary, adjust your rating. Please do so before the 27th of March.

---

### Meta-Review · Area_Chair_Exnb · 2024-04-01

**Recommendation:** Accept (Poster)
**Confidence:** 4

**Metareview:**

The paper explores adversarial attacks which impact less represented groups in the datasets. In the initial phase the reviewers agreed on the topic’s relevance and interesting concerns, but raised some concerns about generalizability of the experiments, and practical ways to alleviate the problems. The authors seem to have largely addressed the concerns, following the reviewers to improve their scores. Given both relevance of the topics, positive discussion and high ratings I would recommend acceptance of this paper.

---

### Decision · Program_Chairs · 2024-04-06

Accept (Oral)